# Self-organization of Plk4 regulates symmetry breaking in centriole duplication

Shohei Yamamoto[1,2] & Daiju Kitagawa[1]

During centriole duplication, a single daughter centriole is formed next to the mother centriole. The molecular mechanism that determines a single duplication site remains a long-standing question. Here, we show that intrinsic self-organization of Plk4 is implicated in symmetry breaking in the process of centriole duplication. We demonstrate that Plk4 has an ability to phase-separate into condensates via an intrinsically disordered linker and that the condensation properties of Plk4 are regulated by autophosphorylation. Consistently, the dissociation dynamics of centriolar Plk4 are controlled by autophosphorylation. We further found that autophosphorylated Plk4 is already distributed as a single focus around the mother centriole before the initiation of procentriole formation, and is subsequently targeted for STIL-HsSAS6 loading. Perturbation of Plk4 self-organization affects the asymmetry of centriolar Plk4 distribution and proper centriole duplication. Overall, we propose that the spatial pattern formation of Plk4 is a determinant of a single duplication site per mother centriole.

---

[1] Graduate School of Pharmaceutical Sciences, The University of Tokyo, Hongo, Tokyo 113-0033, Japan. [2] Department of Biological Science, Graduate School of Science, The University of Tokyo, Hongo, Tokyo 113-0033, Japan. Correspondence and requests for materials should be addressed to D.K. (email: dkitagawa@mol.f.u-tokyo.ac.jp)

The centriole is a nonmembranous organelle that is crucial for centrosome assembly and cilia/flagella formation[1]. Centrioles are duplicated once per cell cycle and only a single daughter centriole is formed next to each preexisting mother centriole[2]. To date, proteins that regulate centriole duplication and their protein–protein interactions have been fairly identified[2,3]. Among these, Polo-like kinase 4 (Plk4) was identified as a master regulator of centriole duplication[4,5]. Loss of Plk4 or inhibition of its kinase activity results in a failure to assemble procentrioles[4–6]. In contrast, overexpression of Plk4 induces multiple daughter centriole formation on a single mother centriole[7,8]. Plk4 phosphorylates SCL-interrupting locus protein (STIL), which subsequently promotes the loading of STIL and spindle assembly abnormal protein 6 (HsSAS6) to the assembly site of a newly formed procentriole[9–13]. Meanwhile, Plk4 autophosphorylates its kinase domain to promote its own kinase activity[14] and its degron motif present in a flexible linker (Linker 1, L1) to enhance its own degradation via the Skp1-Cullin-F-box protein (SCF) and β-transducin repeat containing protein (β-Trcp) ubiquitin ligase-proteasome pathway[15–18]. Despite accumulating studies, the mechanism by which the local amount and/or the kinase activity of Plk4 at mother centrioles are coordinated to ensure formation of a single daughter centriole on a mother centriole remains elusive.

In this study, we identified a molecular property of Plk4 that self-assembles into condensates via Linker 1 domain in vitro and in cells. Autophosphorylation of Plk4 changes condensation properties of Plk4 in both cases. The regulated self-assembly tunes accumulation and dissociation of Plk4 at centrioles. Our data suggest that Plk4 generates spatial patterning around the mother centriole through the self-organization, to duplicate a single procentriole. In fact, we show that the occurrence of Plk4 activation is already detectable before STIL-HsSAS6 loading into a single site around the mother centriole. Thus, these findings imply that self-condensation and autonomous activation of Plk4 are implicated in the determination of a site for STIL-HsSAS6 loading to trigger daughter centriole formation.

## Results

**Autophosphorylation regulates in vitro solubility of Plk4.** To obtain insights into the molecular basis of centriole duplication, we investigated the molecular properties of human Plk4. While performing in vitro biochemical experiments, we unexpectedly found that a kinase-dead (KD) mutant of a purified Plk4 fragment (Kinase + Linker 1) was highly insoluble compared with wild-type (WT) (Fig. 1a, b, Supplementary Fig. 1a). Based on this observation, we speculated that Plk4 may modulate its solubility by autophosphorylation, because Plk4 is known to autophosphorylate its kinase domain and L1[14,16,18–21]. Recent studies have indicated that protein solubility can be regulated by multiple phosphorylation within the low-complexity region (LCR)[22–24]. Because human Plk4 contains an LCR in its disordered L1, we substituted some of the known and putative autophosphorylation sites in or around the LCR with alanine (10A) to generate a nonphosphorylatable mutant or with glutamic acid (10E) to generate a phosphomimetic mutant, as described previously[16,19] (Fig. 1c). In line with this, the phosphorylation-induced mobility shift of Plk4 WT fragments was largely suppressed by the 10A mutation, suggesting that the target residues in WT fragments were phosphorylated in vitro (Supplementary Fig. 1b).

To assess the solubility of purified Plk4 fragments in vitro, we performed a spin-down assay and showed that Plk4 WT fragments were mostly soluble, whereas the KD mutant was largely insoluble (Fig. 1d, Supplementary Fig. 1c and 1d). As expected, the 10A mutant of the Plk4 fragment was mainly

detected in the insoluble fraction, unlike the WT. In contrast, adding the 10E mutation in the KD mutant (KD10E) improved the solubility of the Plk4 fragments compared with the KD mutant. We also monitored the light scattering of purified GST-Plk4(Kinase + L1)-His$_6$ in solution as an indicator of protein aggregation under thermal control. When the temperature was increased, all the Plk4 fragments increased their light scattering but exhibited the onset of scattering at different temperatures (Fig. 1e, Supplementary Fig. 1e). Compared with the WT, the KD and 10A mutants started scattering at lower temperatures. Adding the 10E mutation to the KD mutant shifted the onset of scattering to a higher temperature, than that of the KD mutant. In addition, dephosphorylation of the WT fragment by adding λ-phosphatase (λPP) significantly increased the scattering over time (Fig. 1g), demonstrating that phosphorylation of Plk4 prevents Plk4 aggregation. Furthermore, we confirmed the aggregation properties of Plk4 in vitro using the Proteostat dye, which fluoresces after being incorporated into insoluble protein aggregates. Similarly, KD, 10A, and λPP-treated WT Plk4 fragments showed considerable protein aggregation compared with WT and the other mutants (Fig. 1f, Supplementary Fig. 1f and 1g). Because it appeared that the 10A and 10E mutations did not completely mimic the KD and constitutively phosphorylated states of Plk4, respectively, it is possible that phosphorylation at other sites may also cooperatively regulate the solubility of that Plk4 proteins or amino-acid substitution might affect the natural self-assembly processes[25,26]. Taken together, we conclude that autophosphorylation of Plk4 regulates not only its protein degradation in cells but also its solubility by modulating its self-aggregating properties (Fig. 1h).

**Autophosphorylation changes condensation properties of Plk4.** It has been shown that some proteins self-assemble into macromolecular structures such as spheres, networks, and fibrils which are linked to their physiological functions[27,28]. To characterize Plk4 self-assembly further, we visualized purified Plk4 fragments (Kinase + L1) and full-length Plk4 by labeling with GFP and mScarlet I, respectively. Intriguingly, we found that the Plk4 fragments (Kinase + L1) form condensates in a crowded environment (8% poly-ethylene glycol (PEG)) (Fig. 2a, Supplementary Fig. 2a–d). Importantly, such structures were barely detectable for the Kinase domain alone, suggesting that L1 domain confers condensation property of the Plk4 fragments. Furthermore, the KD and the nonphosphorylatable mutant of GFP-Plk4 fragments (Kinase + L1) exhibited condensates in a lower (2–4%) PEG environment, compared with the WT (Fig. 2a, Supplementary Fig. 2d). This was also true for the full length Plk4 WT and KD (Fig. 2b, Supplementary Fig. 2e–g). These results suggest that Plk4 has an intrinsic ability to self-assemble into condensates and that the condensation properties are regulated by autophosphorylation in vitro.

We next asked whether Plk4 also self-assembles in cells. We overexpressed the Plk4 (Kinase + L1) fragments in HeLa cells and found that depending on the degree of its expression level, the Plk4 fragments self-assemble into spherical structures (Fig. 2c, Supplementary Fig. 3a). In these structures, known Plk4 interactors such as Cep152 and STIL, were hardly detectable, suggesting that this system is suitable for specific validation of Plk4 self-assembly in cells (Supplementary Fig. 3e). Importantly, consistent with in vitro observation of the purified Plk4 proteins, expression of the kinase domain alone did not induce such structures, demonstrating the dependency on the L1 domain for self-condensation (Fig. 2c, Supplementary Fig. 3a). The Plk4 condensates fused and exhibited dynamic turnover (Fig. 2d, e). We also observed similar behavior of the cytoplasmic condensates

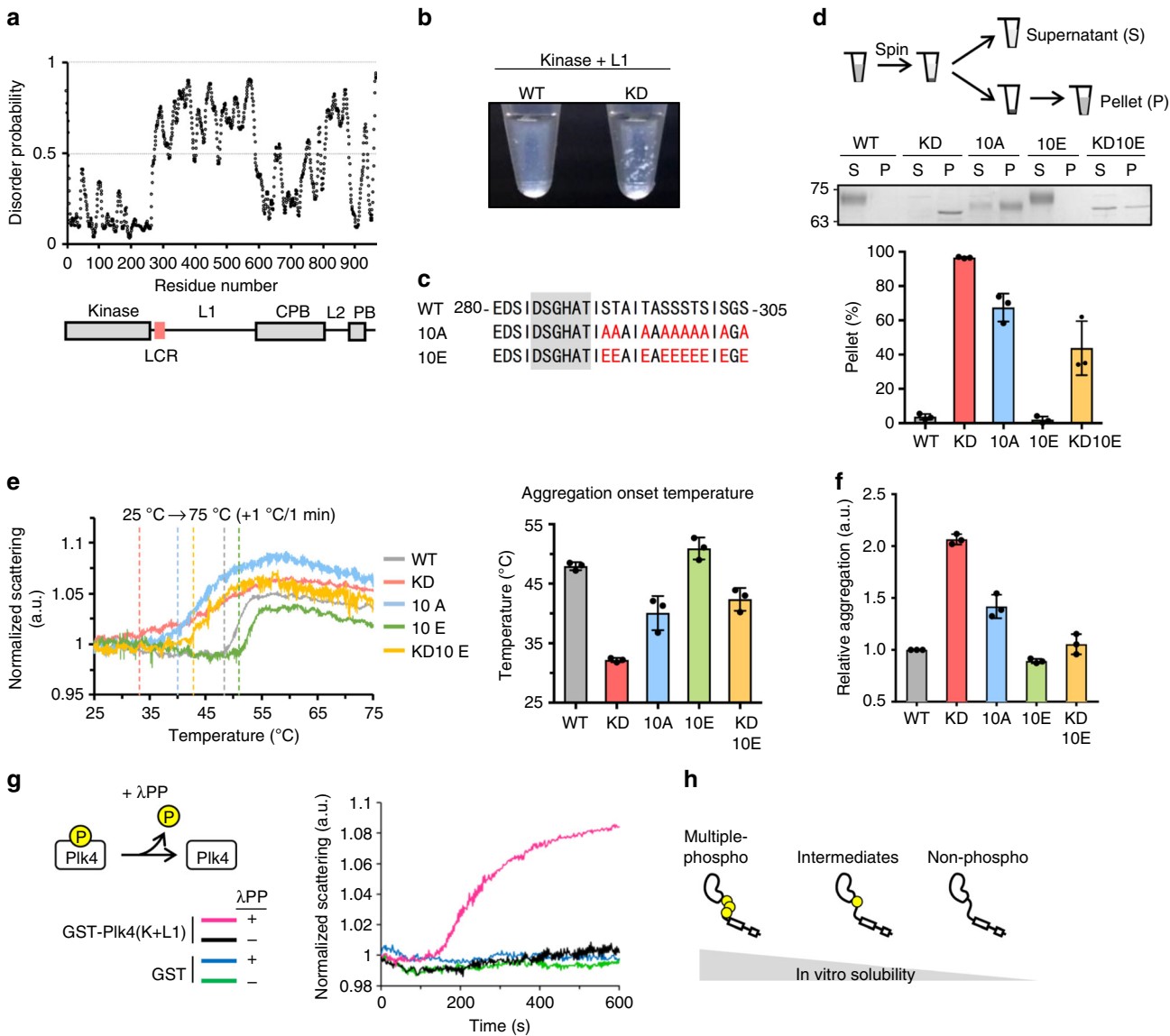

**Fig. 1** Plk4 modulates its solubility by autophosphorylation in vitro. **a** Disorder prediction and schematic of human Plk4 domains. Kinase kinase domain, L1/L2 Linker 1/2, LCR low complexity region, CPB Cryptic Polo-box, PB Polo-box. **b** Image of 500 nM Plk4 (Kinase + L1)-His₆ fragments after GST-tag cleavage by PreScission protease. This truncated form was used for in vitro experiments to directly examine the function of L1 in Plk4 condensation. **c** Amino acid sequence of low complexity region (288–303 a.a.) and its neighboring region of human Plk4. Red letters, mutation sites; Gray background, degron motif. **d** Spin-down assay. After GST-tag cleavage, 500 nM Plk4 (Kinase + L1)-His₆ solution was centrifuged and separated into supernatant (S) and pellet (P) fractions. Representative gel stained with CBB and the quantification are shown. Graph represents mean percentages ± SD of the pellet fraction from three independent experiments. NaCl, 500 mM. **e** Measurement of the light scattering of GST-Plk4 (Kinase + L1)-His₆ (100 μg/ml) under thermal control. Left: Representative data of three independent experiments. Values were normalized to the scattering at 25 °C. Dotted vertical lines indicate each aggregation onset temperature. Right: Aggregation onset temperature. Graph shows mean temperature ± SD from three independent experiments. **f** Measurement of protein aggregation of 300 nM GST-Plk4 (Kinase + L1)-His₆ by Proteostat aggregation assay. Fluorescence values were normalized to the WT. Graph shows mean ± SD from three independent experiments. **g** Dephosphorylation assay. GST-His₆ and GST-Plk4(Kinase + L1)-His₆ (100 μg/ml) were incubated with λPP at 30 °C for the indicated time and each light scattering was measured. Representative data of three independent experiments. **h** Schematic of in vitro solubility of Plk4. Source data are provided as a Source Data file

induced by overexpressing full length Plk4 (Supplementary Fig. 3b–d). Thus, these results suggest that Plk4 has an ability to assemble into liquid-like condensates in cells. We also noticed that centrinone-treatment which specifically inhibits Plk4 kinase activity induced irregular, non-spherical assemblies (Fig. 2c), implying some changes of condensation properties of Plk4 depending on its kinase activity.

To reveal the significance of autophosphorylation for Plk4 condensation in cells, we performed fluorescence recovery after photobleaching (FRAP) analysis with HeLa cells expressing GFP-Plk4. In control cells, cytoplasmic condensates of the Plk4 fragments (Kinase + L1) showed dynamic turnover (Fig. 2e). Intriguingly, we found that inhibition of Plk4 kinase activity with centrinone, but not inhibition of its protein degradation by MG132 (an inhibitor of proteasomes) or MLN4924 (an inhibitor of the NEDD8-activating enzyme that activates SCF-cullin-RING ubiquitin ligase) treatment, significantly impaired the Plk4 turnover (Fig. 2e), although all of these inhibitors similarly increased the amount of endogenous Plk4 at centrioles (Supplementary Fig. 4a). Importantly, the dynamics of full length Plk4 at

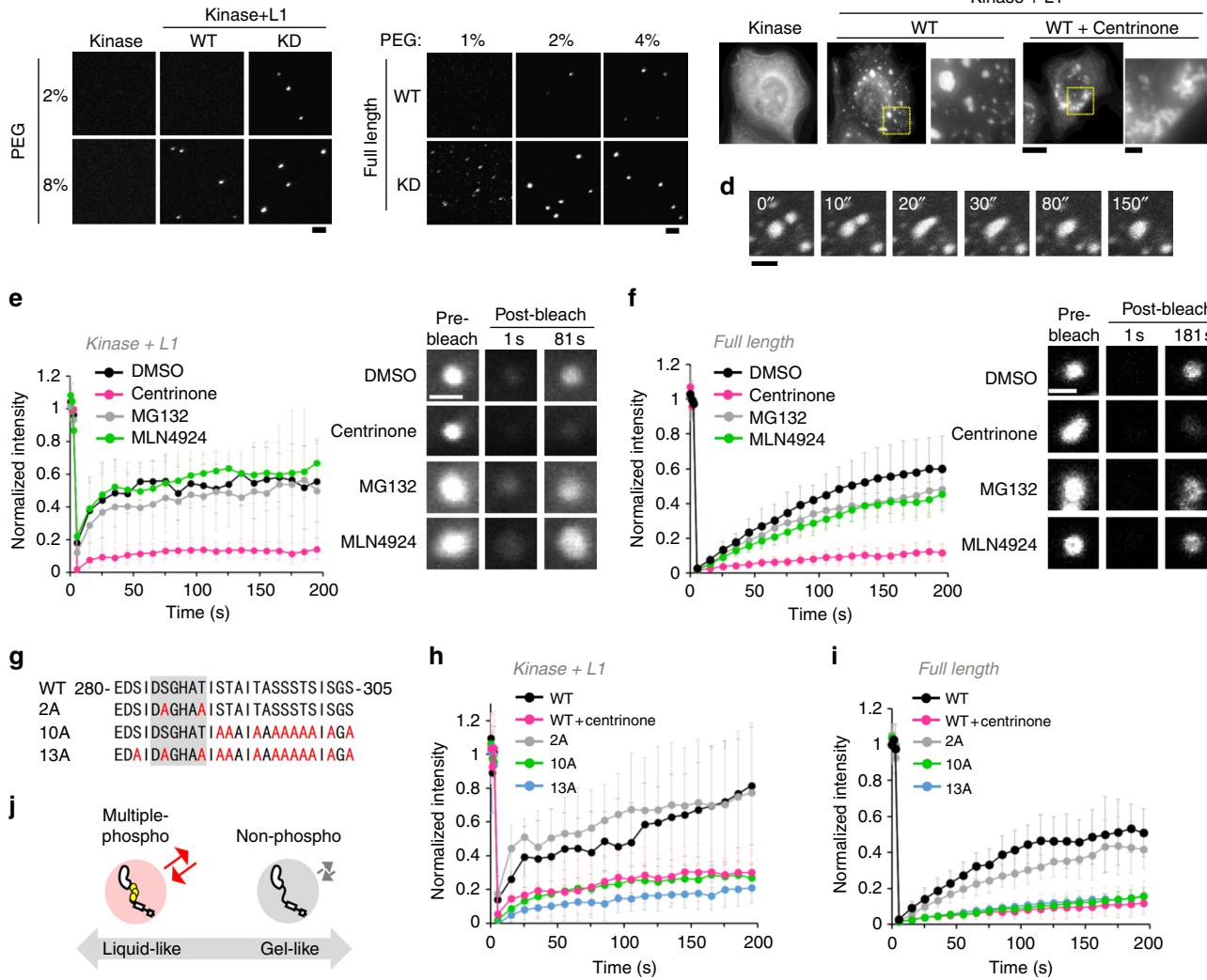

**Fig. 2** Condensation properties of Plk4 are regulated by autophosphorylation. **a** Representative images of 70 nM GFP-Plk4 (Kinase) and GFP-Plk4 (Kinase + L1) in the presence of PEG. Scale bar, 2 μm. **b** Representative images of 40 nM mScarlet I-Plk4 (Full length) in the presence of PEG. Scale bar, 2 μm. **c** HeLa cells expressing GFP-Plk4 (Kinase) and GFP-Plk4 (Kinase + L1). Cells were treated with DMSO or 100 nM Centrinone (20 h). Scale bar, 10 μm; inset, 2 μm. **d** Fusion event of GFP-Plk4 (Kinase + L1)-induced cytoplasmic condensates. Scale bar, 2 μm. **e**, **f** FRAP analysis of cytoplasmic GFP-Plk4 (Kinase + L1) condensates (**e**) and centriolar GFP-Plk4 (Full length) (**f**) in HeLa cells (n = 10 and 15 cells, respectively). Cells were treated with DMSO, 100 nM centrinone, 10 μM MG132 or 10 μM MLN4924 for 5–6 h respectively. **g** Amino acid sequence of mutation sites in Plk4. Red letters, mutation sites; Gray background, degron motif. **h**, **i** FRAP analysis of cytoplasmic GFP-Plk4 (Kinase + L1) mutant condensates (**h**) and centriolar GFP-Plk4 (Full length) mutants (**i**) in HeLa cells (n = 15 cells for each experiment). For **e**, **f**, **h**, **i**, intensities were normalized with the average of three pre-bleach signals. Graph shows mean ± SD from two (**e**) or three (**f**, **h**, **i**) independent experiments. Scale bar, 1 μm. **j** Schematic of the material properties of Plk4 regulated by autophosphorylation. Source data are provided as a Source Data file

centrioles exhibited similar dynamics depending on its kinase activity (Fig. 2f), suggesting that Plk4 condenses at centrioles likewise. From these results, we assumed that centriolar dynamics of condensed Plk4 are regulated by its kinase activity rather than by local protein degradation. Next, we tested whether introducing nonphosphorylatable mutations to Plk4 affects the dynamics of Plk4 condensates. Consistent with the effects of the inhibitor treatment, KD, 10A, and 13A mutations, but not 2A mutation (in the DSG phospho degron motif[16,20]), attenuated Plk4 turnover both in the cytoplasmic condensates and at centrioles (Fig. 2g–i, Supplementary Fig. 4b and 4c). We also confirmed the differences of condensation properties between Plk4 mutants using the plasma membrane tethering assay (Supplementary Fig. 5a–c). Together with the in vitro data, these results indicate that the dynamics of condensed Plk4 changes depending on its

autophosphorylation state (Fig. 2j). We speculate that the autophosphorylation-regulated condensation properties of Plk4 may be conserved across species, because we found that *Drosophila* Plk4 (Dm Plk4) exhibited similar behaviors in HeLa cells (Supplementary Fig. 6a–e).

**Condensation of Plk4 can regulate centriole copy number.** We further analyzed the correlation between Plk4 condensation properties and centriolar Plk4 dynamics. We made several alanine mutants of Plk4 (6A, 7A, 8A1, 8A2, 9A) and found that by changing the number of alanine mutations, the dynamics of cytoplasmic Plk4 condensates was gradually altered (Fig. 3a, b). Importantly, centriolar Plk4 dynamics was also gradually suppressed in concert with an increase in the number of alanine

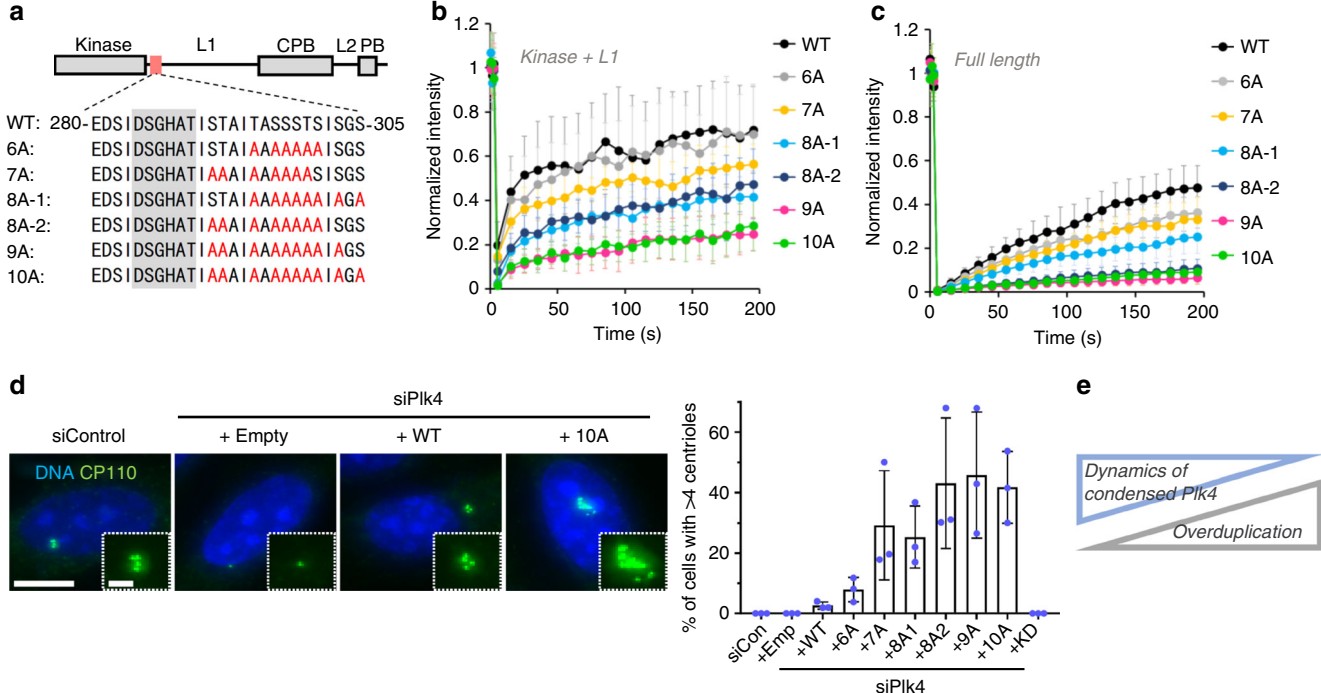

**Fig. 3** Condensation properties of Plk4 regulate centriolar copy number. **a** Amino acid sequence of mutation sites in Plk4. Red letters, mutation sites; Gray background, degron motif. **b** FRAP analysis of cytoplasmic GFP-Plk4 (Kinase + L1) condensates (**b**) and centriolar GFP-Plk4 (Full length) (**c**) in HeLa cells ($n = 12$ and 13 cells, respectively). For **b** and **c**, intensities were normalized with the average of three pre-bleach signals. Graph shows mean ± SD from two (**b**) or three (**c**) independent experiments. **d** Centriole over-duplication assay. Plk4 (Full length)-3xFLAG was exogenously expressed under the CMV mutant promoter in siPlk4-treated HeLa cells. Cells were transfected with siRNA (24 h) and then with plasmid (48 h). Cells were stained with anti-CP110 antibodies as a centriole marker. Scale bar, 10 μm; inset, 2 μm. Percentages of cells which have >4 CP110 foci were calculated. Graph represents mean percentages ± SD (siControl $n = 146$, siPlk4 + Emp $n = 159$, siPlk4 + WT $n = 153$, siPlk4 + 6A $n = 153$, siPlk4 + 7A $n = 166$, siPlk4 + 8A1 $n = 155$, siPlk4 + 8A2 $n = 148$, siPlk4 + 9A $n = 151$, siPlk4 + 10A $n = 157$, siPlk4 + KD $n = 151$ cells) from three independent experiments. **e** Summary of the correlation between the condensed Plk4 dynamics and frequencies of centriole overduplication. Source data are provided as a Source Data file

mutations (Fig. 3c). These results further support the notion that centriolar Plk4 dynamics is critically dependent on its condensation properties.

We next investigated whether the regulated condensation of Plk4 is involved in proper centriole duplication. In human cells, exogenous overexpression of Plk4 under a strong promoter such as CMV (cytomegalovirus) induces centriole overduplication[5,16]. To express the Plk4 mutants in HeLa cells at levels comparable to endogenous expression, we utilized the CMV mutant promoter (see Methods). As expected, in this condition, expression of Plk4-3xFLAG WT rarely induced centriole overduplication (Fig. 3d, Supplementary Fig. 7a and 7b). In contrast, expression of the Plk4 alanine mutants led to centriole overduplication in the same condition. Intriguingly, by introducing alanine mutations to be less dynamic and promote condensation, frequencies of centriole overduplication were significantly enhanced (Fig. 3d, e). This close correlation strongly suggests that condensation properties of Plk4 are implicated in the regulation of centriole copy number and also that the condensation state of Plk4 at centrioles must be appropriately controlled by its autophosphorylation. Overall, we conclude that autophosphorylation of Plk4 regulates not only its degradation but also its condensation and dynamics at centrioles, which is most likely involved in tight control of the centriole copy number.

**Biased distribution of autophosphorylated Plk4.** Our results raised the possibility that the regulated condensation of Plk4 is a key factor for determining a single daughter duplication site. To further investigate how Plk4 condensation regulates its centriolar

localization and centriole duplication, we first analyzed the spatial patterning of endogenous Plk4 during centriole duplication. Recently, we reported that Plk4 forms a biased ring in G1 phase before centriolar loading of STIL-HsSAS6[9,29]. We then confirmed that in G1 phase, Plk4 formed a ring-like structure with an intense focus (Fig. 4a, b). Thereafter, Plk4 changed its localization pattern to a single focus that colocalized with STIL and HsSAS6 in G1/S phase, as reported previously[9,30] (Fig. 4b). Next, to monitor autophosphorylation of Plk4 in cells, we generated the specific antibody against phosphorylated serine 305 (pS305) of Plk4 in this study[19,29] (Supplementary Fig. 8a–d). After mitosis, as Plk4 signals at centrioles were increased, Plk4pS305 signals started to be detected and simultaneously increased before the loading of STIL-HsSAS6 (Fig. 4c, Supplementary Fig. 9a–c). Strikingly, structural illumination microscopy (SIM) revealed that even in cells with a ring-like distribution of Plk4, Plk4pS305 appeared as a single focus around mother centrioles (Fig. 4b). In addition, the Plk4pS305 signal overlapped with the intense focus of Plk4 rings (Fig. 4b, Supplementary Fig. 10a). Consistent with this, even in cells in which HsSAS6 had not yet been loaded to the centrioles, the Plk4pS305 signals were already patterned as a single focus around mother centrioles (Fig. 4d). After HsSAS6 loading, the Plk4pS305 signals colocalized with HsSAS6 as a focus (Fig. 4d). Thus, these results demonstrate that, before procentriole formation, distribution of autophosphorylated Plk4 is already biased around mother centrioles, which could provide the assembly site of procentrioles. Furthermore, even in cells depleted of STIL and HsSAS6 (siSTIL and siHsSAS6-treated), the Plk4pS305 signal was localized as a single focus around mother

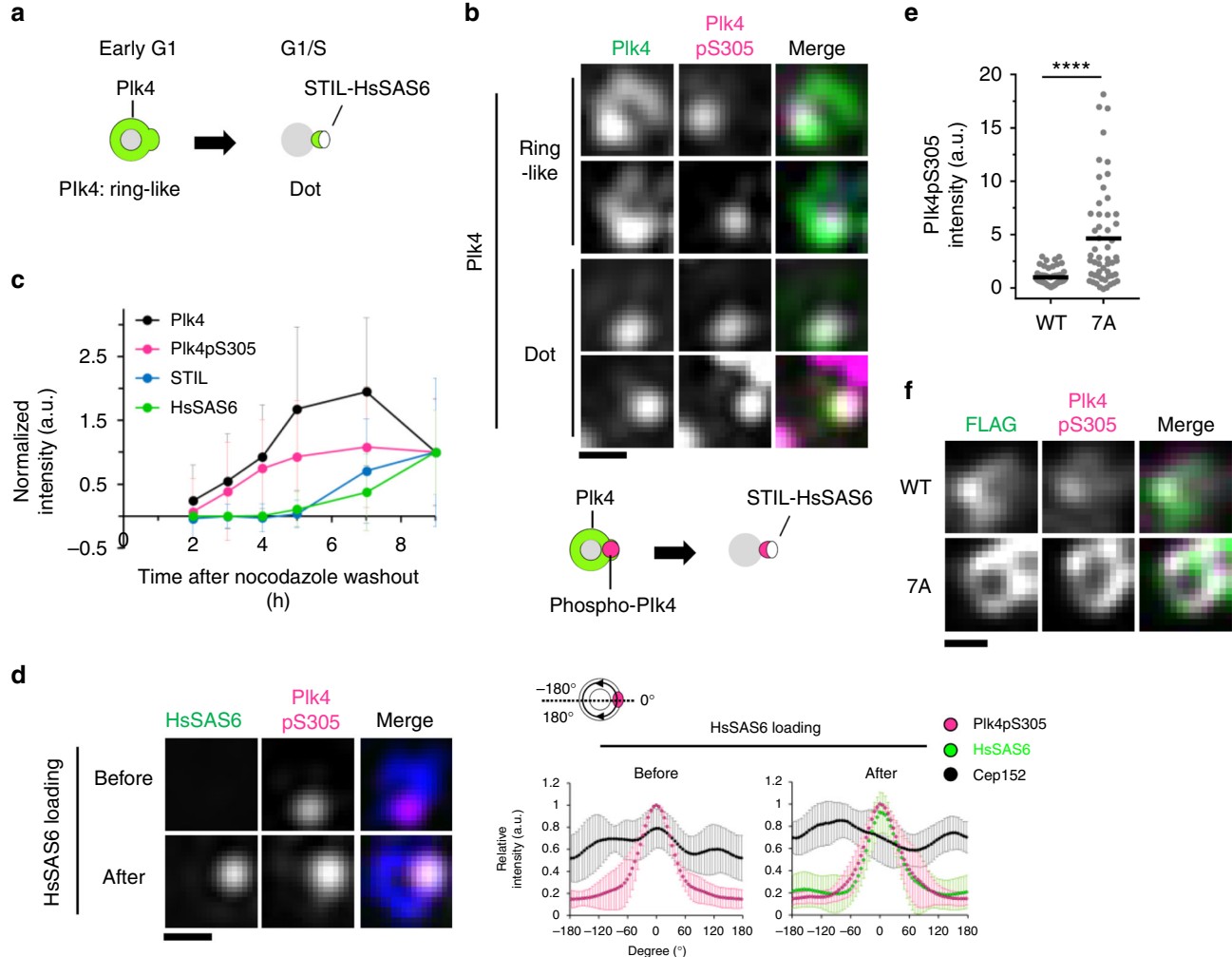

**Fig. 4** A single focus of autophosphorylated Plk4 is generated prior to STIL-HsSAS6 loading. **a** Schematic of Plk4 distribution around mother centrioles in centriole duplication. **b** 3D-SIM images of centrioles immunostained with the indicated antibodies in HeLa cells. Cep152 was costained and used for a marker of mother centriole wall. Scale bar, 0.3 μm. **c** Quantification of centriolar Plk4, Plk4pS305, STIL, and HsSAS6 intensities during G1 phase. Cells were synchronized with thymidine and nocodazole. Mitotic cells were collected and released from nocodazole. Intensities were normalized with the average intensity at 9 h after nocodazole washout. Graph shows mean ± SD of centriolar intensities (For each time point (2, 3, 4, 5, 7, and 9 h), Plk4 $n = 94$, 92, 99, 106, 108, and 102 centrioles, Plk4pS305 $n = 94$, 92, 99, 106, 108, and 102 centrioles, STIL $n = 72$, 91, 97, 92, 98, and 103 centrioles, HsSAS6 $n = 99$, 100, 102, 107, 103, and 102 centrioles) from two independent experiments. See also Supplementary Fig. 9. **d** 3D-SIM images of centrioles immunostained with antibodies against HsSAS6, Plk4pS305, and Cep152 in HeLa cells. Relative intensities of each protein were quantified along Cep152 ring and maximum peaks of Plk4pS305 intensity were set to 0°. Intensities were normalized to each maximum intensity. Graph shows mean intensities ± SD ($n = 11$ and 10 centrioles, respectively). Scale bar, 0.3 μm. **e, f** HeLa cells were transfected with siPlk4 and siHsSAS6 (24 h) and then transfected with plasmid (26 h) expressing Plk4-3xFLAG under the CMV mutant promoter. After plasmid transfection, cells were arrested at G1 phase with Lovastatin (10 μM, 20 h). **e** Quantification of centriolar Plk4pS305 intensities in cells expressing Plk4-3xFLAG WT or 7A. Black bars indicate average intensities ($n = 54$ and 53 centrioles, respectively). Intensities were normalized with the average intensity of the WT. Representative data of two experiments. ****$p < 0.0001$ (Mann–Whitney $U$ test). **f** Representative images of centrioles immunostained with the indicated antibodies. Images were obtained by TCS SP8 HSR system with deconvolution. Scale bar, 0.3 μm. Source data are provided as a Source Data file

centrioles (Supplementary Fig. 10b–e), further confirming that the occurrence of Plk4 phosphorylation is independent of the presence of STIL-HsSAS6 and is likely dependent on the intrinsic properties of Plk4.

To examine whether condensation properties of Plk4 are involved in the autophosphorylation state of Plk4 at centrioles, we utilized Plk4 7A mutant that is less-dynamic, due to its promoted condensation property, and recognized by anti-Plk4pS305 antibodies (Fig. 3a–c). Intriguingly, cells expressing Plk4-3xFLAG 7A under the CMV mutant promoter led to higher amount of Plk4pS305 at centrioles, frequently with a uniform ring structure around mother centrioles (Fig. 4e, f). This result suggests that

condensation properties of Plk4 are involved in spatial organization of autophosphorylated Plk4 at centrioles. We also found that overexpression of Plk4 WT (under CMV promoter) induced a uniform ring-like structure of Plk4pS305 around the mother centriole, suggesting that Plk4 over-condensation induced by its overexpression promoted autonomous activation on the mother centriole wall (Supplementary Fig. 10e). Thus, these data together support the notion that self-condensation of endogenous Plk4 should be properly regulated to ensure a single focus of autophosphorylated Plk4 around mother centrioles, which presumably limits STIL-HsSAS6 loading to the single assembly site for procentrioles.

**Autonomous activation of Plk4 drives STIL-HsSAS6 loading.** Our results thus far demonstrated that Plk4 is autophosphorylated before STIL-HsSAS6 loading, raising the possibility that autonomous activation of Plk4, as a consequence of Plk4 condensation, drives centriolar loading of STIL-HsSAS6 for the initiation of procentriole formation. To address this, we monitored the time course of Plk4 autophosphorylation and STIL-HsSAS6 loading in cells transiently treated with centrinone at G1 phase. We first inhibited the kinase activity of Plk4 by centrinone treatment in G1-arrested HeLa cells and subsequently released the inhibition by washout of centrinone (Fig. 5a). We found that transient centrinone treatment was sufficient to remove the Plk4pS305 signal from centrioles, whereas Plk4 itself significantly accumulated at centrioles (Fig. 5b, c, Supplementary Fig. 11a–e). As previously reported[11,29], centrinone treatment induced displacement of STIL-HsSAS6 from centrioles, indicating that Plk4 kinase activity is required for centriolar localization of the STIL-HsSAS6 complex (Fig. 5b, c, Supplementary Fig. 11a–e). After washout of centrinone, accumulation of Plk4pS305 and STIL-HsSAS6 signals at centrioles began almost simultaneously and increased gradually in most cells (Fig. 5b, c, Supplementary Fig. 11a–e). Importantly, even in HsSAS6-depleted cells (siHsSAS6-treated), the autophosphorylated Plk4 (pS305) signal was increased after washout of centrinone, as observed in control cells (siControl-treated) (Fig. 5b–d, Supplementary Fig. 11a–e), indicating that autophosphorylation of Plk4 occurs in Plk4 condensates independently of centriolar loading of the STIL-HsSAS6 complex. In contrast, as reported previously[31], depletion of HsSAS6 suppressed centriolar loading of STIL (Fig. 5c). Unexpectedly, in this condition, the thickened Plk4 ring did not return to the original state after washout of centrinone (Supplementary Fig. 11e). As a possible reason, we assume that once completely-inactivated Plk4 proteins assemble into less dynamic aggregates upon centrinone treatment, these aggregates trap autophosphorylated Plk4 in the structures without rapid dissociation (Supplementary Fig. 8d). Taken together, our results suggest that (1) autonomous activation of Plk4 within the Plk4 ring occurs in G1 phase, (2) Plk4 activation is mediated by its condensation properties, and (3) Plk4 activation subsequently drives centriolar loading of the STIL-HsSAS6 complex by phosphorylating STIL.

## Discussion

In this study, we demonstrated that Plk4 has an ability to self-assemble into dynamic or less dynamic condensates depending on its autophosphorylation state and that condensation properties of Plk4 are implicated in the regulation of centriole copy number. In our model (Fig. 6), we assume that increasing amount of centriolar Plk4 from late mitosis to late G1 phase is a trigger for its autonomous condensation and activation[14,32]. We found that autophosphorylated Plk4 is spatially distributed as a single focus around mother centrioles, prior to procentriole formation. From these observations, we propose that Plk4 generates a bias of kinase active Plk4 around the mother centriole through condensation-mediated self-organization, which provides the single site for the recruitment of STIL-HsSAS6.

How does Plk4 generate a single focus with its active form? Based on our findings, we hypothesize that the intrinsic self-organization properties of Plk4 are responsible for its spatial distribution. In our model (1), active and inactive Plk4 species that have different diffusion coefficients interact to generate the biased spatial pattern of Plk4 through trans-autophosphorylation[33]. In this model, inactive Plk4 stably self-assembles onto centrioles, while high concentration of Plk4 promotes autonomous activation of Plk4 through trans-autophosphorylation[14]. Activated Plk4 trans-autophosphorylates

inactive Plk4 to become active state[14] and inhibits stable self-assembly of inactive Plk4. The interaction mode of these two components is similar to Turing's reaction-diffusion model[34,35] or its analog, the lateral inhibition model. In addition, the condensation properties of Plk4 may help to generate the biased distribution of Plk4, if the condensation of Plk4 protects Plk4 from degradation/dissociation as seen for several proteins[23,36]. In another model (2), we speculate that active Plk4 proteins are phase-separated into liquid-like droplets around mother centrioles and they eventually form a single and large droplet of active Plk4 through coalescence or Ostwald ripening[37,38], although it remains unclear whether Plk4 can move along the centriole wall. As observed in other nonmembranous organelles[39,40], surface tension of Plk4 droplet may work as a key factor shaping asymmetric structure of Plk4 around mother centrioles. In addition, it is possible that surrounding pericentriolar material (PCM) that is suggested to be a selective condensate[28], may trap Plk4 condensates to regulate fusion, diffusion, and condensation of Plk4. We also assume that these two mechanisms in models (1) and (2) may cooperatively work for symmetry breaking of Plk4. To investigate and improve our model further, other approaches including in vitro reconstitution and super-resolution live imaging should be applied.

Autophosphorylation of Plk4 triggers its own degradation via the SCF ubiquitin ligase-proteasome pathway[15,16,18,20]. Indeed, we observed that inhibition of SCF ubiquitin ligase or proteasome increased the amount of Plk4 at centrioles. However, our FRAP analysis suggests that centriolar Plk4 dynamics are predominantly regulated by self-condensation properties via autophosphorylation rather than by protein degradation; i.e., by dissociation of Plk4 from centrioles. Because it is debatable whether ubiquitin-proteasome machineries can directly ubiquitinate and degrade centriolar Plk4 in a PCM-crowded centrosome, we speculate that the degradation machinery may mainly control the cytoplasmic pool of Plk4, whereas the amount of centriolar Plk4 is controlled by autophosphorylation-regulated condensation and dissociation. In addition, we reason that autophosphorylation-dependent dissociation/diffusion of Plk4 from centrioles could increase fidelity by restricting centriole duplication via more rapid displacement of excess Plk4 from centrioles. In our experiments, the multiple alanine mutations of Plk4 induced centriole overduplication (Fig. 3d, Supplementary Fig. 7). Given that previous studies have suggested that multiple phosphorylation within neighboring regions of the DSG degron somehow enhance Plk4 degradation[16,20], it is unclear whether centriole overduplication induced by expression of the alanine mutants of Plk4 results from defects in the degradation of Plk4 or defects in the regulated self-assembly. Because it has recently been suggested that self-aggregation protects the protein from degradation[23,36], it is possible that degradation of Plk4 is also suppressed by its own aggregation in its nonphosphorylated state. Thus, future investigations are required to distinguish between the effects of autophosphorylation on Plk4 degradation and dissociation/diffusion.

Recent studies have revealed that the intrinsically disordered region (IDR) and LCR act in protein condensation[41,42]. Moreover, it has been shown that protein condensation can be regulated by post-translational modifications[22–24,36,43,44]. We demonstrated that Plk4 self-condenses via an IDR. The Plk4 condensation is regulated by autophosphorylation within the LCR. Thus, our results suggest that Plk4 has a common feature on the regulation of protein condensation. Because protein condensates anchor and concentrate the interactors into the structure to allow its function[28,45], condensed Plk4 could function for selective accumulation of STIL-HsSAS6 to trigger formation of procentrioles. In fact, we found that Plk4 condensates selectively concentrate STIL both in vitro and in cells (Supplementary

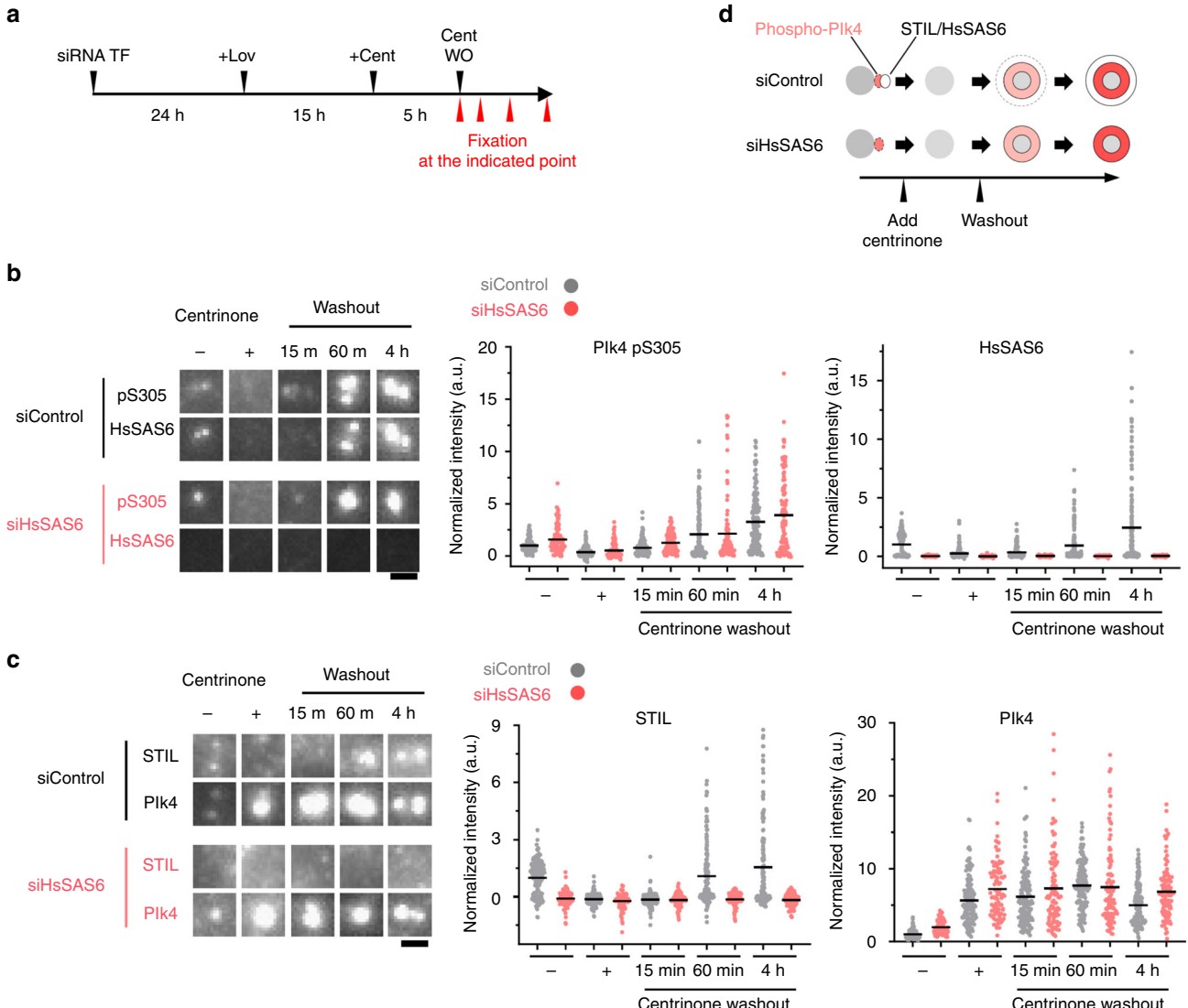

**Fig. 5** Autonomous activation of Plk4 that stems from Plk4 self-condensation drives centriolar loading of STIL-HsSAS6. **a** Scheme of the experimental condition. G1-arrested HeLa cells (10 μM Lovastatin (Lov) (see also Supplementary Fig. 11)) were treated with 100 nM centrinone (Cent) and then washed out (WO) the centrinone. Cells were continuously treated with Lovastatin after centrinone washout. **b, c** Representative images of immunostained HsSAS6 and Plk4pS305 (**b**), STIL and Plk4 (**c**) in the indicated condition. Scale bar, 1 μm. Quantification data show centriolar Plk4pS305, HsSAS6, STIL, and Plk4 intensities in the indicated condition. Gray and pink dots represent the data from siControl and siHsSAS6-treated cells, respectively. Black bars show mean values (for Plk4pS305 and HsSAS6: siControl $n = 166, 161, 171, 154,$ and 160 centrioles, siHsSAS6 $n = 102, 105, 118, 100,$ and 111 centrioles) (for STIL and Plk4: siControl $n = 152, 141, 138, 144,$ and 138 centrioles, siHsSAS6 $n = 109, 96, 104, 105,$ and 102 centrioles) from two independent experiments. Intensities were normalized to the average intensity of the siControl Centrinone (−) condition. **d** Schematic summary of the correlation between centriolar phospho-Plk4 (pS305) and STIL-HsSAS6 intensity. Source data are provided as a Source Data file

Figs. 2h and 3f). Therefore, we propose that Plk4 condensation may be a key factor for procentriole formation through concentrating major centriole components.

It has been reported that the concentration of HsSAS6 is estimated as ~several μM at centrosomes, while ~80 nM in the cytoplasm in human culture cells (early S phase)[46]. Because the relative molecular number of Plk4 at centrosomes is estimated as half of the number of HsSAS6 molecules[47], we speculate that centrosomal Plk4 concentration may be comparable level to HsSAS6 concentration. In addition, the number of Plk4 molecules in cells was quantified as $3670 \pm 620$ molecules/cell in human culture cells[48], suggesting that the cytoplasmic concentration is about 1.5 nM (when cell volume is calculated as 4000 μm³). In our in vitro experiments, Plk4 did not exhibit condensates in its lower concentration (1–5 nM) mimicking cytoplasmic

concentration, while condensation was observed above 10 nM (Supplementary Fig. 2f). Thus, although further investigation including other approaches will be needed, especially in the cellular context, we assume that Plk4 can efficiently undergo condensation through its local concentration at centrioles, but not in the cytoplasm at its endogenous expression level. We think that this concentration-dependent regulation of Plk4 condensation is important for restricting ectopic centriole biogenesis in the cytoplasm.

It has been known that overexpression of STIL-HsSAS6 also promotes centriole overduplication[49]. In our model, we assume the following possibilities to explain the phenomenon. (i) We demonstrated the presence of a single focus of activated Plk4 by immunostaining of Plk4pS305. However, there is a possibility that Plk4 is not completely inactive state but partially active at other

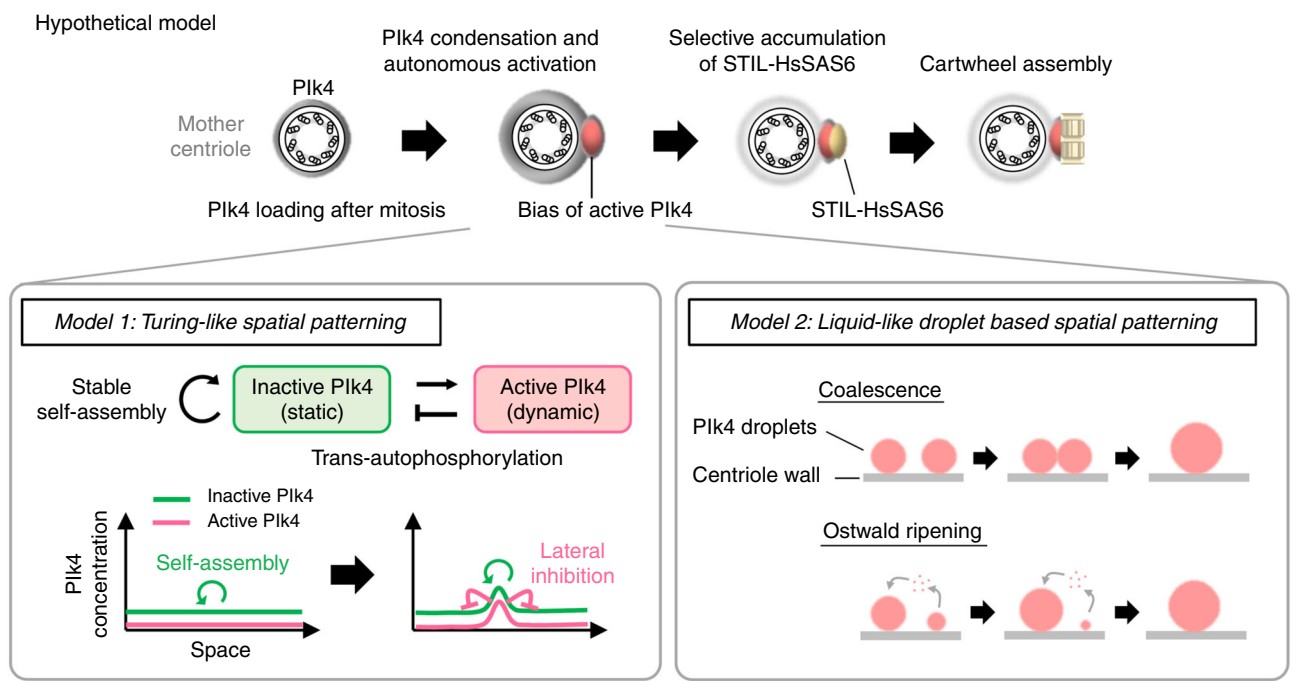

**Fig. 6** Hypothetical model: self-organization of Plk4 directs a single daughter centriole formation per mother centriole. Schematic illustration of hypothetical model. Increasing centriolar Plk4 concentration after mitosis drives condensation and autonomous activation of Plk4 prior to STIL-HsSAS6 loading. Self-organization property of Plk4 driven by the regulated condensation induces spatial pattern formation of autophosphorylated Plk4, which results in a bias formation of activated Plk4. The single focus is a possible target site for STIL-HsSAS6 loading. *Model 1*: active and inactive Plk4 species that have different diffusion coefficients interact with and generate biased distribution of Plk4 around the mother centriole through trans-autophosphorylation. Inactive Plk4 stably self-assembles onto centrioles, while high concentration of Plk4 promotes autonomous activation of Plk4 through trans-autophosphorylation[14]. Activated Plk4 trans-autophosphorylates inactive Plk4 to become active state[14] and suppresses stable self-assembly of inactive Plk4. The interaction mode of these two components is similar to Turing's reaction-diffusion model or its analog, the lateral inhibition model. *Model 2*: active Plk4 proteins are phase-separated into liquid-like droplets around mother centrioles and they eventually form a single and large droplet of active Plk4 through coalescence or Ostwald ripening. Surface tension of Plk4 droplet may work as a key force shaping asymmetric structure of Plk4 around mother centrioles. These two models may cooperatively work for spatial pattern formation of Plk4

sites around the mother centriole. Overexpression of STIL may increase stochastic interaction between STIL and partially activated Plk4 around mother centrioles, resulting in the overloading of STIL-HsSAS6. In human cells, HsSAS6 is required for the loading of STIL at centrioles (Fig. 5c)[31]. Therefore, we speculate that overexpression of HsSAS6 somehow promotes centriolar loading of STIL, resulting in the ectopic complex formation between active Plk4 and STIL and then centriole overduplication. (ii) It has been reported that the binding of STIL to Plk4 promotes further activation of Plk4 (cytosolic and/or around mother centrioles)[11,29]. That may also result in the overloading of active Plk4 around the mother centriole, leading to overloading of STIL-HsSAS6 and centriole overduplication.

Although our study mainly focused on Plk4 properties before centriolar loading of STIL-HsSAS6, it remains to be elucidated how STIL-unbound population of Plk4 is disappeared from mother centrioles after STIL-HsSAS6 loading. It has been reported that overexpression of STIL promotes degradation of Plk4[11,29]. We thus speculate that whereas increasing expression of STIL in G1/S phase limits centriolar loading of Plk4 through the degradation, a stably condensed Plk4 focus incorporating STIL-HsSAS6 complex at the procentriole assembly site might avoid its degradation. It is also possible that the activation and dissociation of STIL-unbound Plk4 are promoted at centrioles via negative feedback regulation, which is mediated by the Plk4-STIL interaction[29]. It will be important to clarify how such transition of Plk4 state is regulated before and after procentriole formation. Furthermore, another important issue to address in the future would be the function of Cep152 and Cep192, known scaffolds

for Plk4, in the regulation of Plk4 status. In particular, it would be conceivable that Cep152 regulates the activity and phosphorylation state of Plk4 on the mother centriole wall and modulates spatial pattern of Plk4 there. Further investigation of the biochemical nature of procentriole components will expand our molecular understanding of centriole duplication.

## Methods

**Plasmid construction.** For recombinant protein expression in *Escherichia coli*, cDNAs encoding human Plk4 full-length and fragments (Kinase: 1–277 a.a., Kinase + L1: 1–585 a.a.), STIL (361–1287 a.a.) were cloned into pGEX 6p-1 (GE Healthcare) and modified by inserting 6× His-tag cDNA sequence. The constructs were further modified by inserting cDNA sequence encoding mScarlet-I or GFP in between PreScission protease recognition site and Plk4 cDNA. For the cleavage of His-tag, cDNA sequence encoding a TEV protease recognition site was further inserted. For over-expression of GFP or mScarlet I-tagged Plk4 in HeLa cells, cDNAs encoding GFP or mScarlet I and human Plk4 or *Drosophila* Plk4 were cloned into pcDNA5/FRT/TO (Thermo Fisher). For weak exogenous expression of GFP-Plk4, the promoter region (between NruI and HindIII sites) of pcDNA5/FRT/TO was replaced with human *Plk4* promoter[50] which was cloned from genomic DNA of HeLa cells using following primers (EcoRV-Plk4 promoter forward: 5′-ATGATATCTGCCCTGTTCCGTCAAGTCT-3′, HindIII-Plk4 promoter reverse: 5′-ATAAGCTTATTTCCAGGCTCTGGCCTGG-3′). For expression of GFP-Plk4-CAAX, pEGFPC1 (CMV promoter) was used and cDNA sequence encoding CAAX motif (derived from pLL7.0: Venus-iLID-CAAX (Addgene # 60411)) was inserted into the vector. 3×FLAG and Plk4-3×FLAG were expressed using pCMV14. For weak exogenous expression of Plk4-3xFLAG in HeLa cells, the CMV immediate early promoter of pCMV14 was partially deleted using primers (CMV mutant forward: 5′-AACGCGGAACTCCATATATGGGC-3′, CMV mutant reverse: 5′- ATGGAGTTCCGCGTTATGGGCGGTAGGCGTGTACG-3′). Plk4 mutants such as kinase-dead (human: D154A, *Drosophila*: D156A), alanine mutants, phospho-mimetic mutants and a ΔL1 mutant (deletion of 278–585 a.a.

region) were generated by using PrimeSTAR mutagenesis basal kit (Takara) and In-Fusion cloning kit (Clontech).

**Protein purification.** *E. coli* strain BL21 gold (DE3) was used for protein expression. Protein expression was induced at 18 °C for 16 h by incubating in LB medium supplemented with 0.3 mM IPTG. Cell pellets were suspended in lysis buffer (50 mM Tris (pH 7.5), 300 mM NaCl, 2 mM MgCl$_2$, 5 mM EDTA, 1 mM DTT, 0.5 mM PMSF, 0.5% TritonX-100) and lysed by lysozyme treatment and sonication. The lysates were then centrifuged at 9000 × *g* for 45 min and the supernatants were collected. The supernatants were incubated with glutathione sepharose beads (GE Healthcare) for 1 h. The beads were washed with 1st wash buffer (50 mM Tris (pH 7.5), 800 mM NaCl, 2 mM MgCl$_2$, 5 mM EDTA, 1 mM DTT, 0.5 mM PMSF, 0.5% TritonX-100) and then washed with pre-elution buffer (50 mM Tris (pH 7.5), 500 mM NaCl, 5 mM β-mercaptoethanol). For purification of GST-His$_6$ or GST-Plk4(Kinase + L1)-His$_6$ fragments, elution was performed in pre-elution buffer supplemented with 20 mM glutathione. For purification of other proteins, elution was performed by incubation with GST-tagged PreScission protease (GE Healthcare) at 4 °C, 3 h or overnight. The eluates were incubated with Ni-Agarose beads (Wako) in binding buffer (50 mM Tris (pH 7.5), 500 mM NaCl, 0.5% TritonX-100, 10 mM Imidazole, 5 mM β-mercaptoethanol) for 1 h. The beads were washed with 2nd wash buffer (50 mM Tris (pH 7.5), 500 mM NaCl, 50 mM Imidazole, 5 mM β-mercaptoethanol) and then proteins were eluted with elution buffer (50 mM Tris (pH 7.5), 500 mM NaCl, 300 mM Imidazole, 5 mM β-mer-captoethanol). The eluates were dialyzed in dialysis buffer (20 mM Tris (pH 7.5), 300 mM or 500 mM NaCl, 1 mM β-mercaptoethanol). The dialyzed samples were treated with His-tagged TEV protease (GenScript) at 4 °C overnight. Then, the TEV protease was removed by using Ni-Agarose beads and His-tag cleaved proteins were collected. Protein concentration was determined by Bradford assay.

**Spin-down assay.** 500 nM of GST-Plk4(Kinase + L1)-His$_6$ fragments were incubated at 30 °C for 1 h in kinase buffer (20 mM Tris (pH 7.5), 300 mM or 500 mM NaCl, 30 μM ATP, 5 mM MgCl$_2$, 1 mM β-mercaptoethanol). The protein samples were incubated with GST-tagged PreScission protease at 4 °C overnight. After the reaction, the samples were centrifuged at 21,500 × *g* for 10 min and the supernatant was collected as supernatant fraction. The pellet was once washed with kinase buffer and then resuspended in kinase buffer (same volume as supernatant fraction) as pellet fraction. The fractions were analyzed by SDS-PAGE (10% poly-acrylamide gel) and CBB staining using SimplyBlue SafeStain (Thermo Fisher). Band intensities were measured using Fiji (NIH). Uncropped images were shown in the Source Data file and Supplementary Fig. 1.

**Detection of protein aggregation.** To detect the light scattering of protein solution, we used Prometheus NT.48 (NanoTemper technologies). 100 μg/ml of GST-His$_6$ and GST-Plk4(Kinase + L1)-His$_6$ fragments in buffer (20 mM Tris (pH 7.5), 500 mM NaCl, 30 μM ATP, 2.5 mM MgCl$_2$, 1 mM β-mercaptoethanol) were loaded into high sensitivity capillaries (NanoTemper Technologies). According to the manufacturer's instruction, the samples were subjected to a thermal ramp (15–95 °C, 1 °C/min) with an excitation power of 100%. Data analysis was performed with the Prometheus PR. ThermControl software (NanoTemper Technologies). Aggregation onset temperature was calculated from first derivative analysis on the software.

To detect the relative amount of protein aggregation, we used PROTEOSTAT protein aggregation assay kit (Enzo, ENZ-51023). 300 nM of protein samples in buffer (20 mM Tris (pH 7.5), 300 mM NaCl, 1 mM β-mercaptoethanol) were supplemented with 0.1 mM MnCl$_2$ and incubated at 37 °C for 1 h with or without lambda phosphatase (NEB, P0753S). According to the manufacturer's instruction, PROTEOSTAT detection reagent was added to each sample on 96-well clear bottom black plate (Corning, 3603). The fluorescence was detected using FilterMaxF5 (Molecular Devices) with excitation wavelength 535 nm and emission wavelength 595 nm (Fig. 1f, Supplementary Fig. 1f) or using FLUOstar OPTIMA microplate reader (BMG Labtech) with excitation wavelength 544 nm and emission wavelength 590 nm (Supplementary Fig. 1g).

**Human cell culture, cell synchronization, and transfection.** HeLa cells were obtained from the ECACC. Cells were cultured in DMEM containing 10% FBS and 1% penicillin/streptomycin at 37 °C in 5% CO$_2$ atmosphere. Cells were arrested in G1 phase by incubating with 10 μM Lovastatin for 20 h. For synchronization in S phase, cells were treated with 6 μM aphidicolin for 24 h.

For the observation of cells during G1 phase progression, cells were once synchronized with 100 μM thymidine for 24 h and then released for 3 h. Cells were then treated with 50 ng/ml nocodazole for 12 h. Mitotic cells were collected by shaking off and released from nocodazole.

Transfection of plasmid DNA and siRNA was performed using Lipofectamine 2000 and Lipofectamine RNAiMAX (Life Technologies), respectively, according to the manufacturer's instruction. Transfected cells were analyzed 48 h after transfection with siRNA and 20, 24, or 48 h after transfection with plasmid DNA.

**RNA interference.** The following siRNAs were used: custom siRNA (Sigma Genosys) against 3′UTR of Plk4 (5′-CTCCTTTCAGACATATAAG-3′); Stealth

siRNA (Life Technologies) against 3′UTR of HsSAS-6 (5′-GAGCUGUUAAAGA CUGGAUACUUUA-3′); custom siRNA (JBios) against 3′UTR of STIL (5′-GTT TAAGGGAAAAGTTATT-3′); Silencer select Negative Control No.1 siRNA (Ambion, 4390843) and Stealth siRNA negative control Low GC Duplex #2 (Invitrogen, 12935110).

**Antibodies.** The following primary antibodies were used; Rabbit polyclonal antibodies against Cep152 (Bethyl Laboratories, A302–480A, IF 1:1000), STIL (Abcam, ab89314, IF 1:50), Cep192 (Bethyl Laboratories, A302–324A, IF 1:1000), CP110 (Proteintech, 12780-1-AP, IF 1:1000), GFP (MBL, 598, IF 1:1000), Plk4 phospho-S305 (IF 1:500, WB 1:2000, the details are described below.) Mouse monoclonal antibodies against Plk4 (Merck Millipore, clone 6H5, MABC544, IF 1:500), HsSAS6 (Santa Cruz Bio-technology, Inc., sc-81431, IF 1:500), Centrin-2 (Merck Millipore, clone 20H5, 04-1624, IF 1:1000), α-tubulin (Sigma, T5168, IF 1:1000), Polyglutamylation Modification (GT335) (AdipoGen, AG-20B-0020-C100, IF 1:1000), PCNA (Santa Cruz Bio-technology, Inc., sc-56, IF 1:1000), GFP (Invitrogen, A11120, IF 1:1000), GST (MBL, M071-3, WB 1:2000), FLAG (Sigma, F1804, IF 1:1000). The following secondary antibodies were used; Alexa Fluor 488 goat anti-mouse IgG (H + L) (Molecular Probes, A11001, IF 1:1000), Alexa Fluor 488 goat anti-rabbit IgG (H + L) (Molecular Probes, A11008, IF 1:1000), Alexa Fluor 594 goat anti-mouse IgG (H + L) (Molecular Probes, A11005, IF 1:1000), Alexa Fluor 568 goat anti-rabbit IgG (H + L) (Molecular Probes, A11011, IF 1:1000), Goat polyclonal antibodies-HRP against mouse IgG (Promega, W402B, WB 1:5000), rabbit IgG (Promega, W401B, WB 1:5000).

Alexa 647-labeled anti-Cep152 (Bethyl Laboratories, A302–480A) and anti-Cep192 (Bethyl laboratories, A302–324A) were generated using Alexa Fluor 647 antibody labeling kit (Molecular Probes, A20186) and used for triple staining.

The Plk4pS305 antibody was generated by immunizing a rabbit with a S305-phosphorylated peptide corresponding to amino acids 301–314 (SISGpSLFDKR RLLC) of human Plk4. The peptides and antiserum were prepared by Eurofins Genomics K.K. (Tokyo, Japan). The antiserum was subjected to affinity-purification using the phosphorylated peptide and then absorbed non-specific antibodies using non-phosphorylated peptide.

**Chemicals.** The following chemicals were used: Centrinone (MedChem Express, HY-18682), MG132 (Wako, 135-18453), MLN4924 (Chemscene, CS-0348), Aphidicolin (Sigma, A0781), Lovastatin (Merck, 438185), Thymidine (Sigma, T1895), and Nocodazole (Wako, 140–08531).

**Imaging of purified proteins.** Fluorescence-labeled proteins were mixed with a buffer containing PEG (20 mM Tris (pH 7.5), 100 mM NaCl, PEG8000 (Promega, V3011) with the indicated concentration) and incubated at RT for 10 min. Only for Supplementary Fig. 2f, 125 mM NaCl was contained in the buffer. The samples were mounted onto slide glasses (Matsunami, S0318) and covered with cover glasses (Matsunami, C015001). Images of condensates were taken using Leica TCS SP8 inverted confocal microscope equipped with a Leica HCX PL APO × 63/1.4 oil CS2 objectives and excitation wavelength 488 or 552 nm. For counting condensates per image (18.49 × 18.49 μm), fluorescence signals above the defined threshold intensity and size were regarded as aggregates and the numbers were measured using Particle analysis in Fiji (NIH).

**Immunostaining and imaging of cells.** HeLa cells cultured on coverslips (Matsunami, C015001) were fixed with cold Methanol at −20 °C for 7 min. Cells were washed with PBS for 5 min three times and incubated in blocking buffer (1% BSA, 0.05% Triton X-100 in PBS) for 30 min. Cells were then incubated with primary antibodies in blocking buffer at 4 °C, overnight and washed with PBS three times, and incubated with secondary antibodies for 1 h at RT. Cells were washed with PBS twice and stained with Hoechst 33258 (DOJINDO) in PBS for 5 min at RT, and then mounted onto slide glasses.

For Figs. 2a–d and 4f, Supplementary Figs. 2, 3d, 5, 6, 8b, 10d, 10f and 11e, Leica TCS SP8 inverted confocal microscope equipped with a Leica HCX PL APO 63×/NA 1.4 oil CS2 objective and excitation wavelength 405, 488, 552 nm was used. Deconvolution of images (Fig. 4f, Supplementary Figs. 10d, 10f and 11e) was performed using Huygens essential software (SVI). For Fig. 5, Supplementary Figs. 4a, 7d–f and 11a–c, DeltaVision personal DV-SoftWorx system (Applied Precision) equipped with an Olympus 60×/NA 1.42 oil objective and a CoolSNAP ES2 CCD camera or EDGE/sCMOS 5.5 camera was used. The images were collected at 0.2 μm z-steps. For Figs. 3d and 4c–e, Supplementary Figs. 3, 7a–c, 8c–d, 9, 10b–c and 11d, Zeiss Axio Imager M2 equipped with a 63×/NA 1.4 Plan-APOCHROMAT oil objective and an AxioCam HRm camera was used. The images were collected at 0.25 μm z-steps. For counting condensates of GFP-Plk4 (Kinase) or (Kinase + L1) in HeLa cells, fluorescence signals 1.8-fold higher than the mean intensity of the cell were regarded as condensates and the number was counted using Particle analysis in Fiji. For counting condensates of GFP-Plk4 (ΔCPB)-CAAX in HeLa cells, images were obtained by z-projection of a plasma membrane region (basal side, 0.4 μm Z-steps × 3). Then, fluorescence signals 3-fold higher than the mean intensity of the plasma membrane region were regarded as condensates. For quantification of fluorescence intensities of centriolar proteins, the images without deconvolution were used and z-stack images were obtained

with maximum intensity projection using Fiji (NIH). Intensities were quantified by drawing a region of interest (encircled with 0.75 μm (Supplementary Fig. 4a), 0.54 μm (Fig. 5b, c, Supplementary Fig. 7f), 0.82 μm (Fig. 4c, Supplementary Figs. 9c and 10b, c), 1.02 μm (Fig. 4e) diameter) around the centriole or the cytoplasmic region using Fiji. The centriolar signal intensity was measured by subtracting the mean intensity of cytoplasmic region from the mean intensity of centriole region.

3D-SIM analysis in Fig. 4b–d and Supplementary Fig. 10e was performed using Nikon N-SIM imaging system equipped with Piezo stage, Apo TIRF 100×/NA 1.49 oil objective, iXon DU-897 EMCCD camera (Andor Technology Ltd.) and excitation wavelength 488, 561, and 640 nm. The images were collected at 0.12 μm z-steps. Reconstruction of SIM images was performed using NIS-Element AR software. Z-stack images were obtained using maximum intensity projection. To quantify fluorescence patterns along mother centriole wall in SIM images, reconstructed images were used and oval profile analysis in Image J (NIH) was performed.

**Fluorescence recovery after photobleaching (FRAP).** For FRAP analysis, HeLa cells were cultured on 35 mm glass-bottom dishes (Greiner-bio-one, #627870). Before FRAP analysis, the culture medium was changed to DMEM containing 10% FBS without phenol red (Sigma).

FRAP analysis was performed using Leica TCS SP8 inverted confocal microscope equipped with a Leica HCX PL APO ×63/1.4 oil CS2 objectives in a chamber in 5% CO$_2$ at 37 °C. The pinhole was adjusted at 2.0 airy units. Single section images were recorded at 1.29 s (pre-bleach) and 5 s (Supplementary Fig. 6d) or 10 s (Fig. 2e–i, Supplementary Fig. 4d–g) (post-bleach) intervals. A region of interest (encircled with 2.54 μm diameter) around the centrosomal GFP signals or around the cytoplasmic condensates was bleached with maximum laser power. Mean intensity values of the centrosomal GFP signals (encircled with 1.66 μm diameter) or the cytoplasmic condensates (encircled with 1.30 μm diameter) were measured using Fiji (NIH) and extracellular signals (Figs. 2f–i and 3c, Supplementary Figs. 4d–g and 6d) or cytoplasmic signals (Figs. 2e–h and 3b) were subtracted as background. Intensities were normalized with the average of three pre-bleach signals. Due to centrosome movements, focus was readjusted during imaging.

**Live cell imaging.** For live cell imaging, cells were seeded onto 35 mm glass-bottom dishes (Greiner-bio-one, #627870) and the culture media were replaced to DMEM containing 10% FBS without phenol red (Sigma). Leica TCS SP8 inverted confocal microscope equipped with a Leica HCX PL APO ×63/1.4 oil CS2 objectives was used and cells were maintained in a chamber in 5% CO$_2$ at 37 °C during the experiments. The pinhole was adjusted at 2.0 airy units. The images were collected at 0.5 μm z-steps and 8–9 sections every 10 s.

**Immunoblotting.** Protein samples were subjected to SDS-PAGE with 8% poly-acrylamide gel and transferred onto Immobilon-P membrane (Millipore). The membrane was incubated in 5% skim milk in PBS and probed with the primary antibodies in 5% BSA in PBS, followed by incubation with their respective HRP-conjugated secondary antibodies. The membrane was washed in PBS containing 0.02% Tween. The signal was detected using Amersham ECL Prime Western blotting detection reagent (GE Healthcare) and a Chemi Doc XRC+ (BioRad). Uncropped images were shown in the Source Data file.

**Protein sequence analysis.** IDRs and LCRs were predicted using PrDOS and SMART, respectively.

**Statistics.** Statistical analyses were performed with GraphPad Prism 7 and 8. Statistical tests, sample sizes and $p$ values are described in each figure legend.

**Reporting summary.** Further information on research design is available in the Nature Research Reporting Summary linked to this article.

## Data availability

The data that support the findings of this study are available from the corresponding author upon request. The source data underlying graphs, gels, and blots are provided as a Source Data file.

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

## Acknowledgements

The authors thank Y. Nozaki and T. Ashikawa for supporting experiments; N. Tokai and M. Takekawa for technical advice of 3D-SIM analysis; S. Hata, M. Ohta, and A. Kimura for fruitful discussion; all the members of Kitagawa laboratory for discussions and critical reading of the manuscript. This work was supported by Grant-in-Aid for Young Scientists (A) and for JSPS Fellows and for Scientific Research on Innovative Area from the Ministry of Education, Science, Sports and Culture of Japan, by Takeda Science Foundation, by Mochida Memorial Foundation.

## Author contributions

S.Y. and D.K. designed the study and the experiments. S.Y. performed all of the experiments. S.Y. and D.K. wrote the manuscript.

## Additional information

**Competing interests:** The authors declare no competing interests.

