## [Peer Review File · Nature Communications]

Editorial Note: *This manuscript has been previously reviewed at another journal that is not operating a transparent peer review scheme. This document only contains reviewer comments and rebuttal letters for versions considered at Nature Communications.*

Reviewers' comments:

Reviewer #1 (Remarks to the Author):

Yamamoto and Kitagawa tackle a long-standing question in cell biology: how does one and only one procentriole form on a mother centriole? Their paper demonstrates that the intrinsic self-association of Plk4 and its regulation by autophosphorylation confines Plk4 into a single liquid-like compartment on the mother centriole. This is a very intriguing proposal, and the paper, as a whole, represents a tremendous breakthrough in the field. The work is well executed and presented. I highly recommend it for publication without any additional experiments.

I do have a few minor issues with their model and interpretation of the data. The points can be addressed by simple changes to the text.

1. Description of Figure 2C. I think the authors should change how they describe the structures they see in the WT + centrinone case. Both liquid-like and solid-like condensates could be described as "amorphous", as it is likely that they contain no long range structure (in the >10 nm range). I would call them irregular, non-spherical assemblies. They could be small condensates that stuck together, but did not fuse (since they are not liquid).

2. I think the model in Figure 6 could be more detailed, in particular to explain how the active Plk4 focus forms. In particular, there are two aspects that require further explanation.

a. Can Plk4 move along the centriole wall to concentrate at one site? Or does Plk4 leave the centriole, then rebind at one location?

b. The authors failed to discuss the concept of Ostwald ripening. Droplet fusion is just one aspect of this. Smaller droplets will dissolve due to an internal pressure (called the Laplace Pressure), which favors growth of droplets larger than defined size (called the critical radius). The end result is that only one droplet will remain. Please see Zwicker, Hyman, and Julicher 2015 (Physical review E)

3. I'm not convinced by the author's proposal of a simple Turing-pattern. First off, a Turing mechanism would require strong positive feedback from active Plk4. This doesn't seem to be the case in the experiments. Active Plk4 actually promotes its own dissociation. Also, a Turing mechanism would require that Inactive Plk4 negatively regulate the formation or localization of active Plk4. But, as far as I'm aware, the data don't show that the inactive form can inhibit the active form.

I think a length-dependent Turing model might be more appropriate. Let's say only surface-exposed Plk4 can be degraded or removed. In this case, the formation of active Plk4 condensates would protect Plk4 from degradation or removal. This would create a positive feedback mechanism to enhance active Plk4 localization in the condensate and discourage Plk4 localization elsewhere. A similar example can be found in Su, Ditlev et al 2015 Science (phase separation of LAT clusters protects them from dephosphorylation, which would otherwise dissolve them).

Reviewer #4 (Remarks to the Author):

The authors were very diligent in addressing the comments and criticisms of original reviewer #2,

in many cases by performing new experiments to address specific points. In particular, the *in vitro* experiments were extended to include full-length Plk4. Based on new results, the authors have revised their model for the role of Plk4 self-association in centriole duplication. They now propose that wild-type Plk4 undergoes self-association through phase separation in the transition from G1 to S phase and that autophosphorylation modulates the structure, material properties and dynamics of the assemblies that form. Disruption of autophosphorylation through genetic or chemical means altered the dynamics (reduced dynamics) and number of dot-like centriolar structures (increased number) in cells. Phase separated and autophosphorylated Plk4, which formed a dot-like structure on the mother centriole, was shown to recruit STIL and HsSAS6, which are known to be required for later stages of daughter centriole formation. The authors' also associated the self-association/phase separation behavior with the N-terminal portion of Plk4, which contains the kinase domain and a long intrinsically disordered region. The revised manuscript deserves serious consideration for publication on *Nat Commun*.

Reviewer #5 (Remarks to the Author):

The revised version of the manuscript by Yamamoto and Kitagawa is now providing clear evidences that the full length Plk4 protein is able to self-aggregate into condensates and that the dynamicity of its condensation properties relies on the status of phosphorylation. Nonetheless, they still do not provide convincing experimental evidences that the condensation properties of Plk4 are physiologically relevant and required at the centriole to concentrate the centriole duplication factors and initiate the procentriole formation. I therefore feel that the authors did not address the main criticism of the referee regarding the requirement of Plk4 self-condensation for centriole duplication.

Major points:

1) The authors now provide clear evidences that Plk4 can assemble into condensates in the cytoplasm or at the plasma membrane when tethered with a tag to this compartment, but only in non-physiological conditions (when Plk4 is overexpressed). The question remains of whether the native Plk4 protein without any tag and expressed at endogenous level in a cell is readily able to condensate, and not only accumulate, at the centriole. In other words, is the concentration of Plk4 protein at the centriole sufficient to initiate condensate formation?

2) The authors claim that "the regulated condensation properties of Plk4 is critical for centriole duplication efficiency" and added new data in Figure 3 and extended data figure 7 to further support their idea. Since the mutant forms of Plk4 are prone to form condensates/aggregates, and since Plk4 condensates can accumulate other centriolar proteins (like Cep152 and STIL, as evidenced in Extended Data Figure 3F), it would have been key to show that the structures observed are readily a bulk of centrioles. If they are, the cells should display extra centrosomes upon mitotic entry, is it the case?

Furthermore, the 2A mutant exhibit a dynamicity of condensation similar to the WT form according to Figure 2I and even more dynamicity in its truncated version (kinase+L1) in Figure 2H. Nonetheless, the 2A mutant is more potent to induce extra centrioles (Extended Data Figure 7C, 7D) and is also more accumulated at the centrioles (Extended Data Figure 7E). Therefore, it remains questionable whether the capacity of the mutant versions of Plk4 to induce extra centrioles is related to their condensation properties per se, and not only to their phosphorylation status.

3) The authors propose that the new data they provide in Figure 4E and 4F further support the notion that changing the condensation status of Plk4 affect the distribution of active Plk4 around the mother centriole. However, active Plk4 (pS305 Plk4) is similarly distributed in a ring-like structure upon expression of the 7A mutant (Figure 4F), the 2A mutant (Extended Data Figure 10F), the WT form (Extended Data Figure 10E) while these three forms clearly display different

dynamics of their condensation properties (Extended Data Figure 4C, Figure 2I and 3C), and therefore of their condensation status.

4) One argument of the authors to relate Plk4 condensation status to procentriole initiation is the capacity of Plk4 condensates to concentrate one of the main effectors of cartwheel formation, the protein STIL. However, in the same Extended Figure 3F, they also show that the protein Cep152, another binding partner of Plk4 that targets it to the centriole is also able to be concentrated by Plk4 condensates. Is it that the non-physiological level of Plk4 required to form these cytoplasmic condensates rather triggers the formation of aggregates that contain many centriolar proteins? I feel that this experiment cannot help to interpret what is happening at the centriole in physiological conditions.

5) In this revised version of the manuscript, the authors have used centrinone to control the phosphorylation and then the condensation status of Plk4 (Extended Figure 8D). If the phosphorylation status is reversible upon drug washout, the condensation status is not, as stated by the authors (Extended Figure 8D and Extended Data Figure 11E). This observation again raises the possibility that these condensates/aggregates form only above physiological level of the Plk4 protein and cannot be degraded in a cellular context.

Comments on points raised by the reviewers

We thank all reviewers of our original manuscript for their critical reading and for their useful and constructive comments (typed in blue), which we addressed in the re-revised version with new data (answers typed in black). Following their suggestions, we conducted new experiments and substantially altered the manuscript, as detailed below.

Referees' comments:

Reviewer #1 (Remarks to the Author):

Yamamoto and Kitagawa tackle a long-standing question in cell biology: how does one and only one procentriole form on a mother centriole? Their paper demonstrates that the intrinsic self-association of Plk4 and its regulation by autophosphorylation confines Plk4 into a single liquid-like compartment on the mother centriole. This is a very intriguing proposal, and the paper, as a whole, represents a tremendous breakthrough in the field. The work is well executed and presented. I highly recommend it for publication without any additional experiments.

>We appreciate this reviewer for very supportive and constructive comments on our study.

I do have a few minor issues with their model and interpretation of the data. The points can be addressed by simple changes to the text.

1. Description of Figure 2C. I think the authors should change how they describe the structures they see in the WT + centrinone case. Both liquid-like and solid-like condensates could be described as “amorphous”, as it is likely that they contain no long range structure (in the >10 nm range). I would call them irregular, non-spherical assemblies. They could be small condensates that stuck together, but did not fuse (since they are not liquid).

>We thank this reviewer for pointing this out, and fully agree with this idea. We therefore changed the description of the structure observed in the case of “WT+Centrinone”, to “irregular, non-spherical assemblies” in the re-revised version of the manuscript (Page 5, line 19).

2. I think the model in Figure 6 could be more detailed, in particular to explain how the active Plk4 focus forms. In particular, there are two aspects that require further explanation.

a. Can Plk4 move along the centriole wall to concentrate at one site? Or does Plk4 leave the

centriole, then rebind at one location?

b. The authors failed to discuss the concept of Ostwald ripening. Droplet fusion is just one aspect of this. Smaller droplets will dissolve due to an internal pressure (called the Laplace Pressure), which favors growth of droplets larger than defined size (called the critical radius). The end result is that only one droplet will remain. Please see Zwicker, Hyman, and Julicher 2015 (Physical review E)

>We appreciate this reviewer for providing us with valuable advices for our hypothetical model. Regarding a, we could not distinguish Plk4 dynamics on the surface of centriole wall thus far, but we think that both cases are possible. Regarding b, this model based on the Ostwald ripening should be included and discussed in our hypothetical model. Accordingly, we mentioned these points in the Discussion part (Figure 6, Page 11, line 1-22).

3. I'm not convinced by the author's proposal of a simple Turing-pattern. First off, a Turing mechanism would require strong positive feedback from active Plk4. This doesn't seem to be the case in the experiments. Active Plk4 actually promotes its own dissociation. Also, a Turing mechanism would require that Inactive Plk4 negatively regulate the formation or localization of active Plk4. But, as far as I'm aware, the data don't show that the inactive form can inhibit the active form.

I think a length-dependent Turing model might be more appropriate. Let's say only surface-exposed Plk4 can be degraded or removed. In this case, the formation of active Plk4 condensates would protect Plk4 from degradation or removal. This would create a positive feedback mechanism to enhance active Plk4 localization in the condensate and discourage Plk4 localization elsewhere. A similar example can be found in Su, Ditlev et al 2015 Science (phase separation of LAT clusters protects them from dephosphorylation, which would otherwise dissolve them).

>We appreciate this reviewer for raising this issue and providing us with great advices. We apologize for our confusing description for the hypothesis of Turing-pattern. We think that "active" Plk4 negatively regulates the assembly of "inactive" Plk4, because it has been reported that trans-autophosphorylation of Plk4 promotes activation of Plk4 (Lopes, C. et al. Dev Cell (2015)). Thus, phosphorylation of "inactive" Plk4 by "active" Plk4 would decrease the population of "inactive" Plk4. We conducted mathematical simulation and found that it is possible to explain how symmetry breaking occurs around mother centrioles based on this molecular property of Plk4 (Takao, D. et al. bioRxiv (2018)). For clarify, we modified the description (Page 11, line 3-12) and cartoon of the model (Figure 6). In addition, in the re-revised version of the manuscript, we discussed this interesting idea from this reviewer about length-dependent regulation of Plk4 dissociation (Page 11, line 10-12).

Reviewer #4 (Remarks to the Author):

The authors were very diligent in addressing the comments and criticisms of original reviewer #2, in many cases by performing new experiments to address specific points. In particular, the in vitro experiments were extended to include full-length Plk4. Based on new results, the authors have revised their model for the role of Plk4 self-association in centriole duplication. They now propose that wild-type Plk4 undergoes self-association through phase separation in the transition from G1 to S phase and that autophosphorylation modulates the structure, material properties and dynamics of the assemblies that form. Disruption of autophosphorylation through genetic or chemical means altered the dynamics (reduced dynamics) and number of dot-like centriolar structures (increased number) in cells. Phase separated and autophosphorylated Plk4, which formed a dot-like structure on the mother centriole, was shown to recruit STIL and HsSAS6, which are known to be required for later stages of daughter centriole formation. The authors also associated the self-association/phase separation behavior with the N-terminal portion of Plk4, which contains the kinase domain and a long intrinsically disordered region. The revised manuscript deserves serious consideration for publication on Nat Commun.

>We appreciate this reviewer for valuable comments on our manuscript and also for supporting our manuscript.

Reviewer #5 (Remarks to the Author):

The revised version of the manuscript by Yamamoto and Kitagawa is now providing clear evidences that the full length Plk4 protein is able to self-aggregate into condensates and that the dynamicity of its condensation properties relies on the status of phosphorylation. Nonetheless, they still do not provide convincing experimental evidences that the condensation properties of Plk4 are physiologically relevant and required at the centriole to concentrate the centriole duplication factors and initiate the procentriole formation. I therefore feel that the authors did not address the main criticism of the referee regarding the requirement of Plk4 self-condensation for centriole duplication.

>We appreciate this reviewer for valuable and constructive comments on our study. As detailed below, we tried to address the concerns with new experiments and substantially altered the manuscript in the re-revised version of the manuscript.

There are technical limitations to demonstrate the condensation of Plk4 at centrioles in physiological condition by using current imaging methods, especially because centriole is a

nanometer scale organelle and dynamically moving in cells. In fact, because of the technical difficulties, many frontier works in this field have mainly conducted *in vitro* reconstitution experiments using purified proteins to show evidences for the condensation of proteins inside cells and the requirement of protein condensation for cellular functions (Su, X. et al. *Science* (2016); Woodruff, J.B. et al. *Cell* (2017); Larson. A.G. et al. *Nature* (2017); Sheu-Gruttadauria, J. and MacRae, I.J. *Cell* (2018); Sabarim B.R. et al. *Science* (2018)). In the re-revised version of the manuscript, to obtain physiological relevance of the condensation properties of Plk4, we re-examined *in vitro* experiments using estimated physiological concentration of Plk4 protein (new data in Extended Data Fig. 2f). We demonstrate that under the estimated concentration of Plk4 protein at centrioles, Plk4 forms condensates, while under the estimated cytoplasmic concentration, Plk4 did not form condensates. Therefore, we speculate that Plk4 has an ability to form condensates at centrioles, but not in cytoplasm. We think that the concentration-dependent regulation of Plk4 condensation is important for preventing ectopic centriole biogenesis in the cytoplasm and for restricted duplication next to the pre-existing mother centrioles.

Although convincing experiments which we can do are technically limited, we found the strong correlation between condensation properties of Plk4 and frequencies of centriole overduplication in cells, under physiological condition (Fig. 3). This experiment was done by expressing the engineered Plk4 mutant proteins at “endogenous levels”. The adequately-expressed Plk4 mutants that harbor various condensation properties revealed strong correlation between their condensation and centriole duplication abilities. Thus, we provide an evidence that Plk4 condensation is a key determinant of centriole duplication efficiency, under physiological condition.

With these evidences that support the condensation property and its function of Plk4 in physiological conditions, we believe that our study on the unique molecular properties of Plk4 will advance our understanding of the mechanisms of centriole duplication as well as the concept of protein condensation in centriole duplication. New data and these points are now described in the re-revised manuscript (Page 13, line 6-18).

References

- Su, X. et al. Phase separation of signaling molecules promotes T cell receptor signal transduction. *Science* (2016)
- Woodruff, J.B. et al. The centrosome is a selective condensate that nucleates microtubules by concentrating tubulin. *Cell* (2017)
- Larson. A.G. et al. Liquid droplet formation by HP1a suggests a role for phase separation in heterochromatin. *Nature* (2017)

Sheu-Gruttadauria, J and MacRae, I.J. Phase transitions in the assembly and Function of Human miRISC. *Cell* (2018)

Sabarim B.R. et al. Coactivator condensation at super-enhancers links phase separation and gene control. *Science* (2018)

Major points:

1) The authors now provide clear evidences that Plk4 can assemble into condensates in the cytoplasm or at the plasma membrane when tethered with a tag to this compartment, but only in non-physiological conditions (when Plk4 is overexpressed). The question remains of whether the native Plk4 protein without any tag and expressed at endogenous level in a cell is readily able to condensate, and not only accumulate, at the centriole. In other words, is the concentration of Plk4 protein at the centriole sufficient to initiate condensate formation?

>We thank this reviewer for raising this important question. As this reviewer mentioned, it remained unclear whether Plk4 (without any tag, at endogenous expression level) physiologically forms condensates at centrioles. However, it is hard to assess protein condensation properties in physiological condition, because super resolution live imaging of dynamically moving nano-scale protein assemblies is almost impossible using currently established techniques. Thus, in the light of many leading works in this field (Su, X. et al. *Science* (2016); Woodruff, J.B. et al. *Cell* (2017); Larson. A.G. et al. *Nature* (2017); Sheu-Gruttadauria, J. and MacRae, I.J. *Cell* (2018); Sabarim B.R. et al. *Science* (2018)), we have performed *in vitro* reconstitution experiments.

In the re-revised manuscript, prompted by the comments from this reviewer, we considered the physiological concentration of Plk4. Although the exact concentration of Plk4 at centrioles is unknown, previous studies measured centrosomal concentration of HsSAS6 and the number of HsSAS6 and STIL molecules at centrosomes (Keller, D. et al. *JCB* (2014); Bauer, M. et al. *EMBO J* (2016)). They showed that centrosomal concentration of HsSAS6 is ~ several μM , while cytoplasmic concentration of HsSAS6 is ~ 80 nM (~100 fold lower than centrosomal HsSAS6) in human culture cells (in early S phase) (Keller, D. et al. *JCB* (2014)). Furthermore, it has been shown that the number of Plk4 molecules at centrosomes is about half of the number of HsSAS6 molecules (Bauer, M. et al. *EMBO J* (2016)). Thus, we speculate that centrosomal concentration of Plk4 is supposed to be several hundreds nM ~ several μM . In addition, it has been reported that the molecular number of Plk4 is 3670 ± 620 molecules/cell in RPE1 cells (Holland AJ et al. *Genes&Dev* (2012)). From these data, we estimated that subcellular concentration of Plk4 is about 1.5 nM (when cell volume is regarded as $4000 \mu\text{m}^3$).

To examine condensation of Plk4 in physiological concentration of Plk4 at centrioles, we re-examined *in vitro* experiments using purified Plk4 protein. We found that Plk4 exhibits condensation at the higher concentration (10 nM ~ 150 nM) mimicking local concentration at centrioles, but did not show condensates at the lower concentration (1 nM ~ 5 nM) mimicking cytoplasmic expression level (new data in Extended Data Fig. 2f). These results suggest the existence of critical concentration to induce Plk4 condensation. Similarly, inside cells, Plk4 forms condensates in the cytoplasm by its overexpression (Extended Data Fig. 3b-c). Taken together, we assume that Plk4 concentrated at centrioles through Cep152-Cep192 binding condenses depending on its local concentration. We think that the concentration-dependent regulation of Plk4 condensation is important for preventing ectopic centriole biogenesis in the cytoplasm and also for restricted duplication next to the pre-existing mother centrioles. In consistent with this notion, importantly, concentration-dependent regulation of protein condensation has been widely recognized (Boeynaems, S. et al. Trends in Cell Biol (2018)), suggesting that Plk4 has a common feature of phase separating proteins. Accordingly, we revised our manuscript with new data and discuss this point in the discussion part (new data in Extended Data Fig. 2f , Page 13, line 6-18).

We think that fluorescence-tagging would not significantly affect Plk4 dynamics for centriole duplication, because many studies have used fluorescence-tagged Plk4 without significant effects (Takao, D. et al. bioRxiv (2018); Aydogan, M.G. et al. JCB (2018); Extended Data Fig7d-e in our manuscript).

2) The authors claim that “the regulated condensation properties of Plk4 is critical for centriole duplication efficiency” and added new data in Figure 3 and extended data figure 7 to further support their idea. Since the mutant forms of Plk4 are prone to form condensates/aggregates, and since Plk4 condensates can accumulate other centriolar proteins (like Cep152 and STIL, as evidenced in Extended Data Figure 3F), it would have been key to show that the structures observed are readily a bulk of centrioles. If they are, the cells should display extra centrosomes upon mitotic entry, is it the case?

>We appreciate this reviewer for raising this important question. Prompted by the comments from this reviewer, we investigated whether Plk4 mutant-induced amplified centrioles act as centrosomes in mitosis. We found that over-amplified centrioles induced by expression of Plk4 mutants associate with spindle microtubules and that multi-polar spindles formation was observed in Plk4 mutant-expressing cells (new data in Extended Data Fig. 7d). Therefore, we conclude that expression of Plk4 mutants induced over-amplification of “centrosomes”. We then added the new data and mentioned these results in the re-revised manuscript (legend of Extended Data Fig.7c, Page 41).

Furthermore, the 2A mutant exhibit a dynamicity of condensation similar to the WT form according to Figure 2I and even more dynamicity in its truncated version (kinase+L1) in Figure 2H. Nonetheless, the 2A mutant is more potent to induce extra centrioles (Extended Data Figure 7C, 7D) and is also more accumulated at the centrioles (Extended Data Figure 7E). Therefore, it remains questionable whether the capacity of the mutant versions of Plk4 to induce extra centrioles is related to their condensation properties per se, and not only to their phosphorylation status.

>We thank this reviewer for pointing this out. It has been known that the 2A mutation in the degron motif suppresses degradation of Plk4 protein by preventing the binding to ubiquitin ligases (Cunha-Ferreira, I. et al. *Current Biology* (2013)) and thus increases cytoplasmic Plk4 concentration (i.e. overexpression) inside cells. Because even in the WT form, Plk4 over-condensates inside cells by its overexpression (Extended Data Fig. 3b-c), we think that the 2A mutation led to over-condensation of Plk4 around centrioles, and thus results in centriole over-duplication (Extended Data Fig. 7d-f). However, as this reviewer mentioned, the dynamicity of the 2A mutant is similar to that of WT because the condensation property of the 2A mutant is not significantly altered. In contrast, importantly, the 10A and 13A mutants which are prone to condense induced robust centriole overduplication even under the weak expression promoter (presumably close to the endogenous expression level), compared with the WT and 2A form (Extended Data Fig. 7d-e), although their amount at centrioles was comparable level to that of the 2A form (Extended Data Fig. 7f). Furthermore, we demonstrated the strong correlation between condensation properties of Plk4 and frequencies of centriole overduplication using the mutants which are more prone to condense (Fig. 3). Thus, from these evidences, we conclude that the regulated condensation property Plk4 is a key factor regulating centriole duplication efficiency. In the re-revised version of the manuscript, we mentioned these points more explicitly (Page 6, line 10-11 and Page 41 (legend on Extended Data Fig. 7d-f)).

3) The authors propose that the new data they provide in Figure 4E and 4F further support the notion that changing the condensation status of Plk4 affect the distribution of active Plk4 around the mother centriole. However, active Plk4 (pS305 Plk4) is similarly distributed in a ring-like structure upon expression of the 7A mutant (Figure 4F), the 2A mutant (Extended Data Figure 10F), the WT form (Extended Data Figure 10E) while these three forms clearly display different dynamics of their condensation properties (Extended Data Figure 4C, Figure 2I and 3C), and therefore of their condensation status.

>We apologize for not having described these experiments more explicitly. Expression of the

Plk4 WT (Extended Data Fig. 10e) and Plk4 mutants (7A in Figure 4f, 2A in Extended Data Fig. 10f) was induced under different promoters. As discussed above, overexpression of Plk4 promotes its over-condensation even in the WT form. In the experiment of “Extended Data Figure 10e”, Plk4 WT overexpression (under CMV promoter) induced ring-like structure of Plk4 pS305, because of over-condensation of Plk4 at centrioles. In contrast, endogenous Plk4 (Fig. 4b) and exogenously expressed WT under the weak expression promoter (presumably close to the endogenous expression level, in Fig. 4f) did not show such ring-like structure of pS305. As for the 2A mutant that was designed as a degradation-defective mutant of Plk4, its condensation property is similar to that of the WT. However, as mentioned above, the total expression level of the 2A (in the degron motif) is higher than the endogenous level. This presumably led to over-condensation of Plk4 and an increase of pS305 signal at centrioles (Extended Data Fig. 10f). In contrast, importantly, as for the 7A mutant, expression of this mutant resulted in significant increase of the pS305 signal as well as Plk4 condensation at centrioles, compared to that of the WT form under weak expression promoter (Fig. 4f). These data support the close correlation between Plk4 condensation and distribution of active Plk4 around the mother centriole. Based on these data, we assume that at the endogenous expression level, Plk4 condensation is properly regulated to generate a bias of activated Plk4. In the re-revised manuscript, we described these points in more detail (Page 9, line 3-4, Page 25 (legend of Extended Data Fig. 4e-f), Page 47(legend of Extended Data Fig. 10e)).

4) One argument of the authors to relate Plk4 condensation status to procentriole initiation is the capacity of Plk4 condensates to concentrate one of the main effectors of cartwheel formation, the protein STIL. However, in the same Extended Figure 3F, they also show that the protein Cep152, another binding partner of Plk4 that targets it to the centriole is also able to be concentrated by Plk4 condensates. Is it that the non-physiological level of Plk4 required to form these cytoplasmic condensates rather triggers the formation of aggregates that contain many centriolar proteins? I feel that this experiment cannot help to interpret what is happening at the centriole in physiological conditions.

>We agree with this reviewer that the cytoplasmic condensates might not reflect what is happening at centrioles in physiological conditions. However, given that condensates of GFP-Plk4 mutant (Kinase+L1) did not incorporate STIL and Cep152 into the structures, recruitment of STIL and Cep152 into the condensates of GFP-Plk4 full length seems to be selective. We speculate that Plk4 condensates have an ability to concentrate Cep152, because Plk4 directly binds to Cep152. However, it remains unknown whether condensed Plk4 concentrates Cep152 at centrioles in physiological conditions. As this reviewer pointed

out, since we are not sure how much the data reflect the physiological role of Plk4 condensates at centrioles, we described this issue in a modest and appropriate manner in the Figure legend and the Discussion part (Page 34 (legend of Extended Data Fig. 3f) and Page 13, line 4-5).

5) In this revised version of the manuscript, the authors have used centrinone to control the phosphorylation and then the condensation status of Plk4 (Extended Figure 8D). If the phosphorylation status is reversible upon drug washout, the condensation status is not, as stated by the authors (Extended Figure 8D and Extended Data Figure 11E). This observation again raises the possibility that these condensates/aggregates form only above physiological level of the Plk4 protein and cannot be degraded in a cellular context.

>We thank this reviewer for raising this issue. Our data demonstrated that centrinone treatment promoted Plk4 aggregation at centrioles. The condition used in this study for centrinone treatment, induced over-aggregation of Plk4 through changing the condensation property into more solid state (Fig. 2, Extended Data Fig. 8D, 11e). As this reviewer mentioned, this local concentration of Plk4 at centrioles is above the physiological level. However, in the re-revised manuscript, we provided an evidence that Plk4 forms condensates *in vitro* at estimated physiological concentration at centrioles (new data in Extended Data Fig. 2f). Again, as far as we know, to prove the condensation property of Plk4 physiologically in cells, it would require STED-level super-resolution live imaging. We believe that it is beyond the scope of this study and should be one of challenging future works on this topic. In the re-revised version of manuscript, we added discussion in more detail on this issue (Page 13, line 6-18).

REVIEWERS' COMMENTS:

Reviewer #5 (Remarks to the Author):

I acknowledge the effort made by the authors to provide new in vitro evidence that Plk4 can self-aggregate into condensates at physiological level, and that such condensation might only occur next to the centriole where Plk4 concentration is high enough. Nonetheless, as discussed by the authors, the data provided in the current manuscript cannot readily prove that Plk4 forms condensates next to the centriole in a cellular context and that such condensation is required for centriole duplication to occur. It is indeed a technical challenge, which is to my sense not beyond the scope of this manuscript if the authors want to claim in the title and abstract that "self-organization of Plk4 regulates symmetry breaking in centriole duplication". The in vitro data are convincingly showing that Plk4 is able to form condensates in a phosphorylation-dependent manner, which is already a step forward in our understanding of the regulation of this key centriole duplication factor. I therefore do not feel that the manuscript is ready for publication in its current state.

Comments on points raised by the reviewer

We thank the reviewer of our original manuscript for critical reading and for useful and constructive comments (typed in blue), which we addressed in the re-revised version of manuscript (answers typed in black). Following his/her suggestions, we substantially altered the manuscript, as detailed below.

REVIEWERS' COMMENTS:

Reviewer #5 (Remarks to the Author):

I acknowledge the effort made by the authors to provide new in vitro evidence that Plk4 can self-aggregate into condensates at physiological level, and that such condensation might only occur next to the centriole where Plk4 concentration is high enough. Nonetheless, as discussed by the authors, the data provided in the current manuscript cannot readily prove that Plk4 forms condensates next to the centriole in a cellular context and that such condensation is required for centriole duplication to occur. It is indeed a technical challenge, which is to my sense not beyond the scope of this manuscript if the authors want to claim in the title and abstract that “self-organization of Plk4 regulates symmetry breaking in centriole duplication”. The in vitro data are convincingly showing that Plk4 is able to form condensates in a phosphorylation-dependent manner, which is already a step forward in our understanding of the regulation of this key centriole duplication factor. I therefore do not feel that the manuscript is ready for publication in its current state.

>We appreciate this reviewer for valuable and constructive comments on our manuscript. To describe our conclusion more properly, we modified our manuscript to tone down our claims as detailed below.

In the abstract (page 1, line 16-17), we toned down our claim (“we demonstrate that intrinsic self-organization of Plk4 underlies symmetry breaking...”) to “we show that intrinsic self-organization of Plk4 is implicated in symmetry breaking...”.

In the Results part, to tone down our claims, we modified a subheading (“Condensation properties of Plk4 control centriole copy number”) to “Condensation of Plk4 can regulate centriole copy number” (Page7, line 24).

We also modified the text as below: Page 8, line 17-18: A description (“condensation

properties of Plk4 are critical to control the copy number of procentrioles”) was modified to “condensation properties of Plk4 are implicated in the regulation of centriole copy number”.

Page 8, line 21-22: We modified a sentence (“... which is most likely involved in tight control of the centriole copy number”) to tone down the conclusion.

In the Discussion part, we modified the text as below:

Page 12, line 4: A description (“the regulated condensation of Plk4 at centrioles is critical for proper centriole duplication”) was modified to “condensation properties of Plk4 are implicated in the regulation of centriole copy number” to tone down our conclusion.

Page 15, line 3-4: We added a sentence (“although further investigation including other approaches will be needed, especially in the cellular context”) to describe our conclusion more properly.